# Optimism and pessimism in optimised replay

**Georgy Antonov**[1,2]*, **Christopher Gagne**[1], **Eran Eldar**[3,4], **Peter Dayan**[1,5]

**1** Max Planck Institute for Biological Cybernetics, Tübingen, Germany, **2** Graduate Training Centre of Neuroscience, International Max Planck Research School, University of Tübingen, Tübingen, Germany, **3** Department of Psychology, Hebrew University of Jerusalem, Jerusalem, Israel, **4** Department of Cognitive and Brain Sciences, Hebrew University of Jerusalem, Jerusalem, Israel, **5** University of Tübingen, Tübingen, Germany

* georgy.antonov@tuebingen.mpg.de

## Abstract

The replay of task-relevant trajectories is known to contribute to memory consolidation and improved task performance. A wide variety of experimental data show that the content of replayed sequences is highly specific and can be modulated by reward as well as other prominent task variables. However, the rules governing the choice of sequences to be replayed still remain poorly understood. One recent theoretical suggestion is that the prioritization of replay experiences in decision-making problems is based on their effect on the choice of action. We show that this implies that subjects should replay sub-optimal actions that they dysfunctionally choose rather than optimal ones, when, by being forgetful, they experience large amounts of uncertainty in their internal models of the world. We use this to account for recent experimental data demonstrating exactly pessimal replay, fitting model parameters to the individual subjects' choices.

**Data Availability Statement:** The code and data used to produce the results and analyses presented in this manuscript are available at https://github.com/geoant1/optimism_and_pessimism.

## Author summary

When animals are asleep or restfully awake, populations of neurons in their brains recapitulate activity associated with extended behaviourally-relevant experiences. This process is called replay, and it has been established for a long time in rodents, and very recently in humans, to be important for good performance in decision-making tasks. The specific experiences which are replayed during those epochs follow highly ordered patterns, but the mechanisms which establish their priority are still not fully understood. One promising theoretical suggestion is that each replay experience is chosen in such a way that the learning that ensues is most helpful for the subsequent performance of the animal. A very recent study reported a surprising result that humans who achieved high performance in a planning task tended to replay actions they found to be sub-optimal, and that this was associated with a useful deprecation of those actions in subsequent performance. In this study, we examine the nature of this pessimized form of replay and show that it is exactly appropriate for forgetful agents. We analyse the role of forgetting for replay choices of our model, and verify our predictions using human subject data.

**Funding:** GA, CG, and PD are funded by the Max Planck Society (https://www.mpg.de/en). PD is also funded by the Alexander von Humboldt Foundation (https://www.humboldt-foundation.de/en/). EE holds the National Institute of Health grants R01MH124092 and R01MH125564 (https://www.nih.gov/), and a United States-Israel Binational Science Foundation grant 2019801 (https://www.bsf.org.il/). The funders had no role in study design, data collection and analysis, decision to publish, or preparation of the manuscript.

**Competing interests:** The authors have declared that no competing interests exist.

**Abbreviations:**

**IF**, **Individual flexibility**. A behavioural index introduced by Eldar *et al.* (2020) [31] which measures the extent to which subjects manage to adapt their choices to the design of the task space whereby optimal choices in a subset of trials are sub-optimal in another set of trials. Higher individual flexibility thus means greater behavioural flexibility according to the task rules. Eldar *et al.* (2020) [31] showed that this measure correlated with other measures of the extent to which subjects had and/or used a model of the task.

**MB**, **Model-based**. An algorithm which makes decisions based on prospective values estimated based on a learnt model of the environment. Decision-making via such algorithms is computationally expensive but the use of a model implies that choices can be revalued or devalued if the agent is informed about changes in the task structure.

**MEG**, **Magnetoencephalography**. A brain imaging technique that registers magnetic fields generated due to specific patterns of neural activity. Recently, this technique has been used to decode with high temporal precision the sequential progression of states during replay events.

**MF**, **Model-free**. An algorithm which makes decisions based on a set of retrospective cached values learnt directly from past experience. Decision-making via such algorithms is computationally cheap; however, the downside is that they do not learn any task structure and are therefore quite stubborn when the task rules change, since the cached values are no longer applicable and need to be re-learnt.

**MI**, **Model-informed**. A hybrid model-free / model-based algorithm which makes decisions based on a set of model-free values which, however, can be altered by information supplied by the algorithm's model of the environment.

**RL**, **Reinforcement learning**. An area of machine learning in which by interacting with an

# Introduction

During periods of quiet restfulness and sleep, when humans and other animals are not actively engaged in calculating or executing the immediate solutions to tasks, the brain is nevertheless not quiet. Rather it entertains a seething foment of activity. The nature of this activity has been most clearly elucidated in the hippocampus of rodents, since decoding the spatial codes reported by large populations of simultaneously recorded place cells [1, 2] reveals ordered patterns. Rodents apparently re-imagine places and trajectories that they recently visited ('replay') [3–6], or might visit in the future ('preplay') [6–13], or are associated with unusually large amounts of reward [14–16]. However, replay is not only associated with the hippocampus; there is also a complex semantic and temporal coupling with dynamical states in the cortex [17–23].

In humans, the patterns of neural engagement during these restful periods have historically been classified in such terms as default mode or task-negative activity [24]. This activity has been of great value in elucidating functional connectivity in the brain [25–27]; however, its information content had for a long time been somewhat obscure. Recently, though, decoding neural signals from magnetoencephalographic (MEG) recordings in specific time periods associated with the solution of carefully designed cognitive tasks, has revealed contentful replay and preplay (for convenience, we will generally refer to both simply as 'replay') that bears some resemblance to the rodent recordings [28–32].

The obvious question is what computational roles, if any, are played by these information-ally-rich signals. It is known that disrupting replay in rodents leads to deficits in a variety of tasks [19, 33–36], and there are various theoretical ideas about its associated functions. Although the notions are not completely accepted, it has been suggested that the brain uses off-line activity to build forms of inverse models—index extension and maintenance in the context of memory consolidation [37], recognition models in the case of unsupervised learning [38], and off-line planning in the context of decision-making [39, 40].

While appealing, these various suggestions concern replay in general, and have not explained the micro-structure of which pattern is replayed when. One particularly promising idea for this in the area of decision-making, suggested by Mattar and Daw (2018) [41], is that granular choices of replay experiences are optimized for off-line planning. The notion, which marries two venerable suggestions in reinforcement learning (RL) [42]: DYNA [39] and prioritized sweeping [43], is that each replayed experience changes the model-free value of an action in order to maximize the utility of the animal's ensuing behaviour. It was shown that the resulting optimal choice of experience balances two forces: need, which quantifies the expected frequency with which the state involved in the experience is encountered, and gain, which quantifies the benefit of the change to the behaviour at that state occasioned by replaying the experience. This idea explained a wealth of replay phenomena in rodents.

Applying these optimizing ideas to humans has been hard, since, until recently the micro-structure of replay in humans had not been assessed. Liu *et al.* (2020) [32] offered one compelling test which well followed Mattar and Daw (2018) [41]. By contrast, in a simple planning task, Eldar *et al.* (2020) [31] showed an unexpected form of efficacious replay in humans that, on the surface, seemed only partially to align with this theory. In this task, subjects varied in the extent to which their decisions reflected the utilisation of a model of how the task was structured. The more model-based (MB) they were, the more they engaged in replay during inter-trial interval periods, in a way that appeared helpful for their behaviour. Strikingly, though, the replay was apparently pessimized–that is, subjects preferred to replay *bad* choices, which were then deprecated in the future.

environment, agents acquire systematic methods of acting (policies) that tend to maximize gains and minimize losses.

In this paper, we consider this characteristic of replay, examining it from the perspective of optimality. We show that favouring bad choices is in fact appropriate in the face of substantial uncertainty about the transition structure of the environment—a form of uncertainty that arises, for instance, from forgetting. Moreover, we consider the costs and benefits of replay on task performance in the light of subjects' subjective, and potentially deviant, knowledge of the task. Although, to be concrete, we focus on the task studied by Eldar *et al.* (2020) [31], the issues we consider are of general importance.

## Results

### Preamble

In the study of Eldar *et al.* (2020) [31], human subjects acted in a carefully designed planning task (Fig 1A). Each state was associated with an image that was normally seen by the subjects. The subjects started each trial in a pseudo-random state and were required to choose a move among the 4 possible directions: up, down, left or right (based on the toroidal connectivity

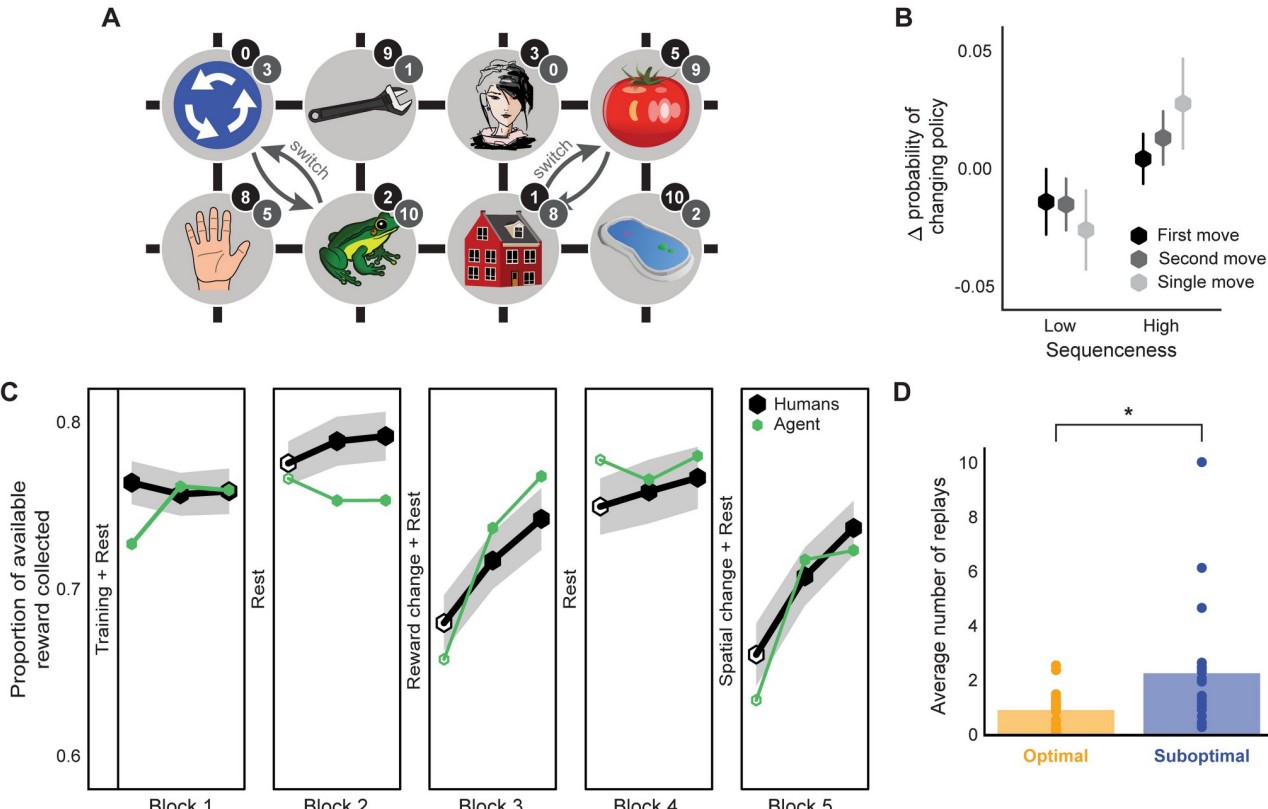

**Fig 1. Task structure and replay modelling.** (A) Structure of the state-space. Numbers in black and grey circles denote the number of reward points associated with that state respectively pre- and post- the reward association change between blocks 2 and 3. Grey arrows show the spatial re-arrangement that took place between blocks 4 and 5. Note that the stimuli images shown here differ from those which the subjects actually saw. (B) Change in the probability of choosing a different move when in the same state as a function of sequenceness of the just-experienced transitions measured from the MEG data in subjects with non-negligible sequenceness ($n = 25$). High sequenceness was defined as above median and low sequenceness as below median. Analysis of correlation between the decoded sequenceness and probability of policy change indicated a significant dependency (Spearman correlation, $M = 0.04$, $SEM = 0.02$, $p = 0.04$, Bootstrap test). Vertical lines show standard error of the mean (SEM). (C) Performance of the human subjects and the agent with parameters fit to the individual subjects. Unfilled hexagons show epochs which contained trials without feedback. Shaded area shows SEM. (D) Pessimism bias in the replay choices of human subjects for which our model predicted sufficient replay ($n = 20$) as reflected in the average number of replays of recent sub-optimal and optimal transitions at the end of each trial (sub-optimal vs optimal, Wilcoxon rank-sum test, $W = 2.49$, $p = 0.013$). ** $p < 0.01$.

shown). In most cases, they were then shown an image associated with the new state according to the chosen move and received a reward associated with that image (this was not displayed; however, the subjects had been extensively taught about the associations between images and reward). Some trials only allowed single moves and others required subjects to make an additional second move which provided a second reward from the final state. In most of these 2-move trials, in order to obtain maximal total reward, the subjects had to select a first move that would have been sub-optimal had it been a 1-move trial.

The subjects were not aware of the spatial arrangement of the state space, and thus had to learn about it by trial and error (which they did in the training phase that preceded the main task, see Methods for details). In order to collect additional data on subjects' knowledge of the state space, no feedback was provided about which state was reached after performing an action in the first 12 trials of blocks two through five. This feedback was provided in the remaining 42 trials to allow ongoing learning in the face, for instance, of forgetting. After two blocks of trials, the subjects were taught a new set of associations between images and rewards; similarly, before the final block they were informed about a pair of re-arrangements in the transition structure of the state space (involving swapping the locations of two pairs of images).

In order to achieve high performance, the subjects had to a) adjust their choices according to whether the trial allowed 1 or 2 moves; and b) adapt to the introduced changes in the environment. Eldar *et al.* (2020) [31] calculated an individual flexibility (IF) index from the former adjustment, and showed that this measure correlated significantly with how well the subjects adapted to the changes in the environment (and with their ability to draw the two different state spaces after performing all the trials, as well as to perform 2-move trials in the absence of feedback about the result of the first move). The more flexible subjects therefore presumably utilised a model of the environment to plan and re-evaluate their choices accurately.

Eldar *et al.* (2020) [31] investigated replay by decoding MEG data to reveal which images (i.e., states) subjects were contemplating during various task epochs. They exploited the so-called sequenceness analysis [28] to show that, in subjects with high IF, the order of contemplation of states in the inter-trial intervals following the outcome of a move revealed the replay of recently visited transitions (as opposed to the less flexible, presumably model-free (MF), subjects); it was notable that the transitions they preferred to replay (as measured by high sequenceness) mostly led to sub-optimal outcomes. Nevertheless, those subjects clearly benefited from what we call pessimized replay, for after the replay of sub-optimal actions they were, correctly, less likely to choose these (Fig 1B) when faced with the same selection of choices later on in the task.

In Eldar *et al.* (2020) [31], and largely following conventional suggestions [44–47], subjects' choices were modelled with a hybrid MF/MB algorithm. This fit the data better than algorithms that relied on either pure MF or MB learning strategies. However, the proposed model did not account for replay and therefore could not explain the preference for particular replays or their effect on choice.

Therefore, to gain further insights into the mechanisms that underpinned replay choices of human subjects (as well as their effect on behaviour), we constructed an agent that made purely MF choices, but whose MF values were adjusted by a form of MB replay [41] that was optimal according to the agent's forgetful model of the task. The agent was therefore able to adapt its decision strategy flexibly by controlling the amount of influence maintained by (subjectively optimal) MB information over MF values. We simulated this agent in the same behavioural task with the free parameters fit to the subjects (Fig 1C), and examined the resulting replay preferences.

## Modelling of subjects' choices in the behavioural task

To model replay in the behavioural task, we used a DYNA-like agent [39] which learns on-line by observing the consequences of its actions, as well as off-line in the inter-trial intervals by means of generative replay (Fig 2A). On-line learning is used to update a set of MF $Q$-values [48] which determine the agent's choices through a softmax policy, as well as to (re)learn a model of the environment (i.e., transition probabilities). During off-line periods, the agent uses its model of the transition structure of the environment to estimate $Q$-values (denoted as $\widehat{Q}^{MB}$), and then evaluates these MB estimates for the potential improvements to MF policy that the agent uses to make decisions. This process of evaluation and improvement is the key difference between our model and that of Eldar *et al.* (2020) [31]: instead of having MB quantities

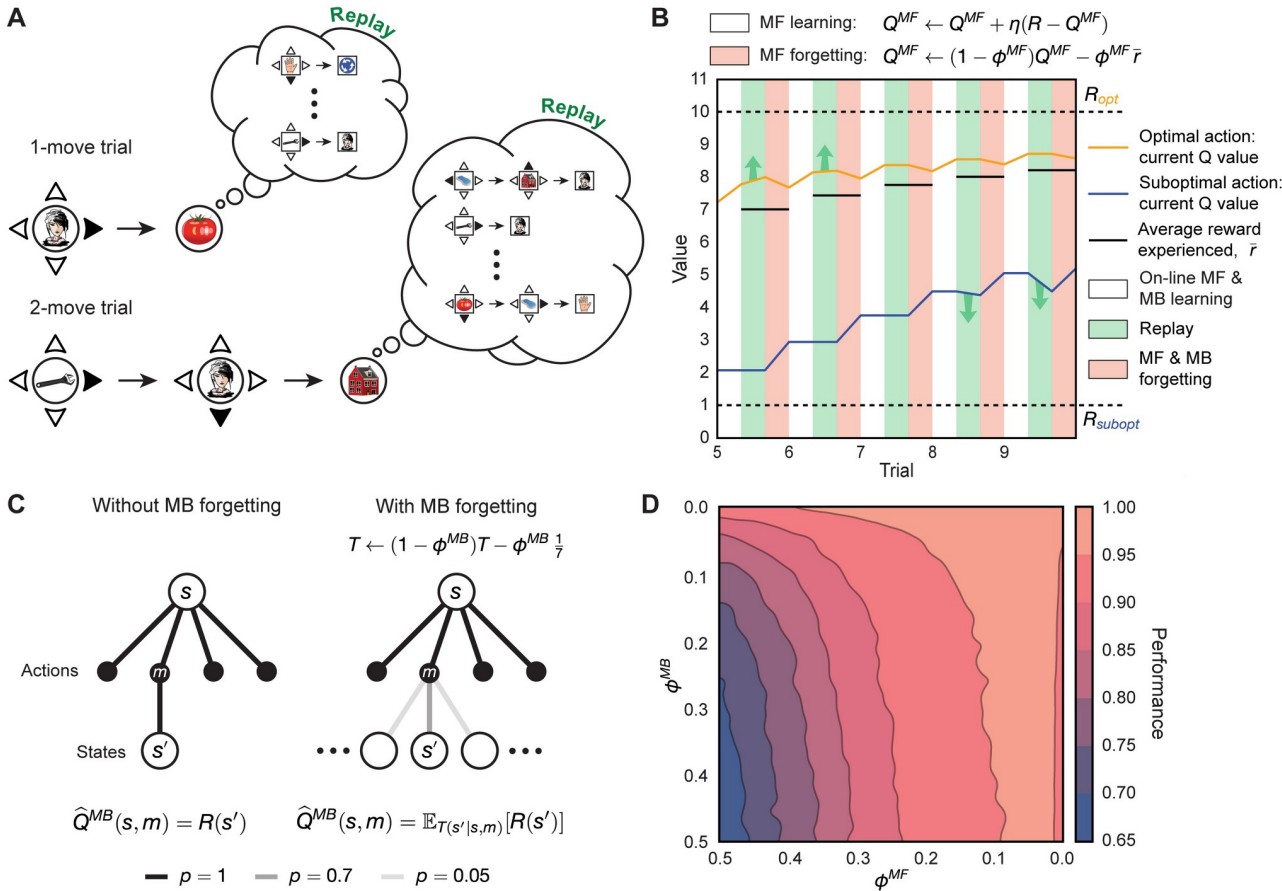

Fig 2. Algorithm description and the effects of replay and forgetting on model performance. (A) Schematic illustration of the algorithm in the behavioural task. Upon completing each trial, the algorithm uses its knowledge of the transition structure of the environment to replay the possible outcomes. Note that in 1-move trials the algorithm replays only single moves, while in 2-move trials it considers both single and coupled moves (thus optimizing this choice). (B) Effect of MF forgetting and replay on MF $Q$-values. After acting and learning on-line towards true reward $R$ (white blocks; controlled by learning rate, $\eta$), the algorithm learns off-line by means of replay (green blocks). Immediately after each replay bout, the algorithm forgets its MF $Q$-values towards the average reward experienced from the beginning of the task (red blocks; controlled by MF forgetting rate, $\phi^{MF}$). Note that after trials 5 and 6, the agent chooses to replay the objectively optimal action, whereas after trials 8 and 9 it replays the objectively sub-optimal action. (C) Left: without MB forgetting, the algorithm's estimate of reward obtained for a given move corresponds to the true reward function. Right: with MB forgetting (controlled by MB forgetting rate, $\phi^{MB}$), the algorithm's estimate of reward becomes an expectation of the reward function under its state-transition model. The state-transition model's probabilities for the transitions are shown as translucent lines. (D) Steady-state performance (proportion of available reward obtained) of the algorithm in the behavioural task as a function of MF forgetting, $\phi^{MF}$, and MB forgetting, $\phi^{MB}$. Note how the agent still achieves high performance with substantial MF forgetting (high $\phi^{MF}$) when its state-transition model accurately represents the transition probabilities (low $\phi^{MB}$).

affecting the agent's choices directly, they only did so by informing optimized replay [41] that provided additional training for the MF values.

Unlike a typical DYNA agent, or indeed the suggestion from Mattar and Daw (2018) [41], our algorithm performs a full model evaluation, and therefore MF $Q$-values are updated in a supervised manner towards the model-generated $\widehat{Q}^{MB}$ values. It is crucial to note that subjects were found by Eldar *et al.* (2020) [31] (and also in our own model fitting) to be notably forgetful. Therefore, although the task is deterministic, replay can help and hurt, since the seemingly omniscient MB updates may in fact be useless, or even worse, harmful. To avoid extensive training of MF values, in the light of the potential harm an inaccurate MB system can accomplish, the agent thus only engages in replay as long as the potential MB updates are estimated to be sufficiently gainful (see below); this is controlled by a replay gain threshold, which was a free parameter that we fit to each individual subject.

Similarly to the best-fitting model in Eldar *et al.* (2020) [31], after experiencing a move, our agent forgets both MF $Q$-values and the state-transition model (for all allowable transitions). However, unlike that account, over the course of forgetting, MF $Q$-values tend towards the average reward the agent has experienced since the beginning of the task (Fig 2B), as opposed to tending towards what was a fixed subject-specific parameter. Insofar as the agent improves over the course of the task, the average reward it obtains increases with each trial. MF $Q$-values for sub-optimal actions, therefore, tend to rise towards this average experienced reward; MF $Q$-values for optimal actions, on the other hand, become devalued, as the agent is prone occasionally to choose sub-optimal actions due to its non-deterministic policy. In other words, because of MF forgetting, the agent forgets how good the optimal actions are and how bad the sub-optimal actions are. Similarly, because of MB forgetting, the agent gradually forgets what is specific about particular transitions, progressively assuming a uniform distribution over the potential states to which it can transition (Fig 2C). The agent therefore becomes uncertain over time about the consequences of actions it rarely experiences.

To disambiguate the contribution of each individual component of our agent to the resulting behaviour, in Fig 3 we compare agents of varying complexity in a simple task involving only two actions (see Methods for details). First, note how agents with optimised replay significantly outperform their non-replaying counterparts (Fig 3A), hence suggesting that replay can indeed improve performance. Fig 3B shows the effects of replay and forgetting on the evolution of MF $Q$-values over the course of learning. Without forgetting (Fig 3B; left), MF $Q$-values quickly converge and remain stable, thus obviating the need for replay (Fig 3C; left) for all but the very first few trials (such an unforgetful agent precisely implements optimised replay as suggested by Mattar and Daw (2018) [41]). By contrast, given MF forgetting, continual learning (implemented, for instance, via additional replay-based off-line training) is required to maintain sound performance (Fig 3A). Importantly, as we further detail below, the extent to which replay can help ameliorate MF forgetting is itself limited by the amount of MB forgetting in the agent's model of the task (notice in Fig 3B that the agent with MB forgetting underestimates MF $Q$-value for the objectively optimal action when compared to the agent without MB forgetting).

The two forgetting mechanisms therefore significantly influence the agent's behaviour—MF forgetting effectively decreases the value of each state by infusing the agent's policy with randomness, whereas MB forgetting confuses the agent with respect to the individual action outcomes. From an optimality perspective, the question is then what and if the agent should replay at all, given the imperfect knowledge of the world and a forgetful MF policy. We find that at high MF forgetting, replay confers a noticeable performance advantage to the agent provided that MB forgetting is mild (as can be seen from the curvature of the contour lines in

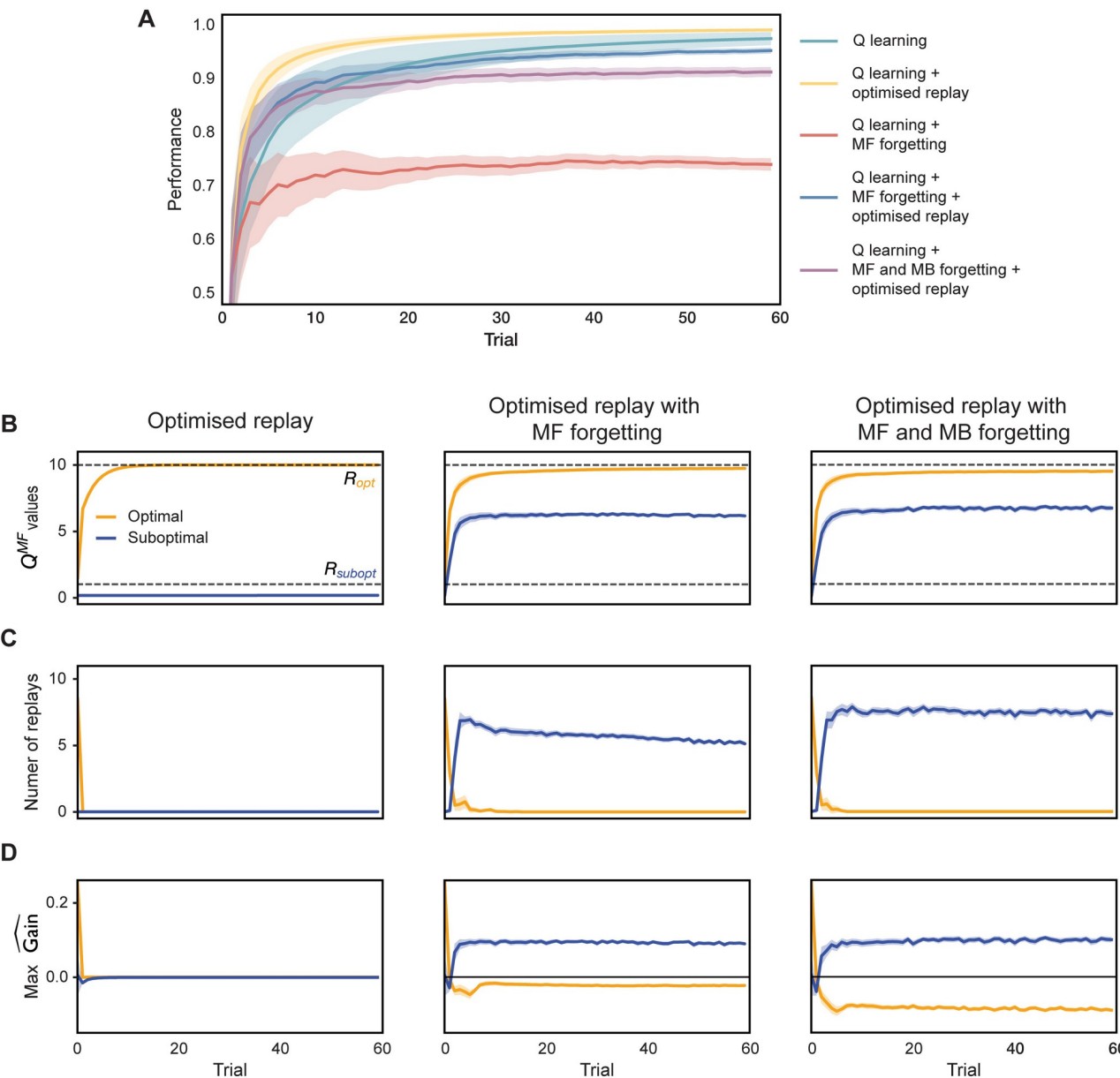

**Fig 3. Incremental model comparison.** (A) Average performance (cumulative proportion of available reward obtained) of agents with varying degree of model complexity. (B) Evolution of MF $Q$-values during learning. Dashed grey lines indicate true reward $R$ for each action. Blue and orange lines indicate MF $Q$-values for the objectively sub-optimal and optimal actions respectively. (C) Number of replays in each trial. (D) Maximal gain for objectively sub-optimal and optimal actions as estimated by the agents in each trial. Shaded areas show 95% confidence intervals.

Fig 2D). This means that, by engaging in replay, the agent is able to (on average) improve its MF policy and increase the obtained reward rate as long as there is little uncertainty about the transition structure.

We then analysed the replay choices of human participants which, according to our model prediction (with the free parameters of our agent fit to data from individual subjects), engaged in replay. This revealed a significant preference to replay actions that led to sub-optimal outcomes (Fig 1D). We therefore considered the parameter regimes in our model that led the

agent to make such pessimal choices, and whether the subjects' apparent preference to replay sub-optimal actions was formally beneficial for improving their policies.

## Exploration of parameter regimes

The analysis in Mattar and Daw (2018) [41] suggests that two critical factors, need and gain, should jointly determine the ordering of replay by which an (in their case, accurate) MB system should train an MF controller. Need quantifies the extent to which a state is expected to be visited according to the agent's policy and transition dynamics of the environment. It is closely related to the successor representation across possible start states, which is itself a prediction of discounted future state occupancies [49]. Heterogeneity in need would come from biases in the initial states on each trial (5 of the 8 were more common; but which 5 changed after blocks 2 and 4) and the contribution of subject's preferences for the first move on 2-move trials. However, we expected the heterogeneity itself to be modest and potentially hard for subjects to track, and so made the approximation that need was the same for all states.

Gain quantifies the expected local benefit at a state from the change to the policy that would be engendered by a replay. Importantly, gain only accrues when the behavioural policy changes. Thus, one reason that the replay of sub-optimal actions is favoured is that replays that strengthen an already apparently optimal action will not be considered very gainful—so to the extent that the agent already chooses the best action, it will have little reason to replay it (such as in Fig 3C and 3D; left). A second reason comes from considering why continuing learning is necessary in this context anyhow—i.e., forgetting of the MF $Q$-values. Since the agent learns on-line as well as off-line, it will have more opportunities to learn about optimal actions without replay. Conversely, the values of sub-optimal actions are forgotten without this compensation, and so can potentially benefit from off-line replay (Fig 3C and 3D; middle and right).

We follow Mattar and Daw (2018) [41] in assuming that the agent computes gain optimally. However, this optimal computation is conducted on the basis of the agent's subjective model of the task, which will be imperfect given forgetfulness. Thus, a choice to replay a particular transition might seem to be suitably gainful, and so selected by the agent, but would actually be deleterious, damaging the agent's ability to collect reward. In this section, we explore this tension.

In Fig 4A, we show the estimated gain for an objectively optimal and sub-optimal action in an example simulation with only two actions available to the agent (see Methods for details). The bar plot in Fig 4B illustrates the 'centering' effect MF forgetting has on sub-optimal MF $Q$-values (which is also evident for forgetful agents in Fig 3B); this effect, however, is not symmetric. As the agent learns the optimal policy, the average reward it experiences becomes increasingly similar to the average reward obtainable from optimal actions in the environment. As a result, there is little room for MF $Q$-values for optimal actions to change through forgetting; on the other hand, MF $Q$-values for sub-optimal actions are forgotten towards what is near to the value of optimal actions to a much more substantial degree. Thus forgetting towards the average reward the agent experiences (provided that it accumulates with learning) is optimistic in a way that favours replay of sub-optimal actions.

Objectively sub-optimal actions that our agent replays therefore mostly have negative temporal difference (TD) errors. Such pessimized replay reminds the agent that sub-optimal actions, according to its model, are actually worse than predicted by the current MF policy.

There is an additional, subtle, aspect of forgetting that decreases both the objective and subjective benefit of replay of what are objectively optimal actions (provided the agent correctly estimates such actions to be optimal). For the agent to benefit from this replay, its state-transition model must be appropriately accurate as to generate MB $Q$-values for optimal actions that

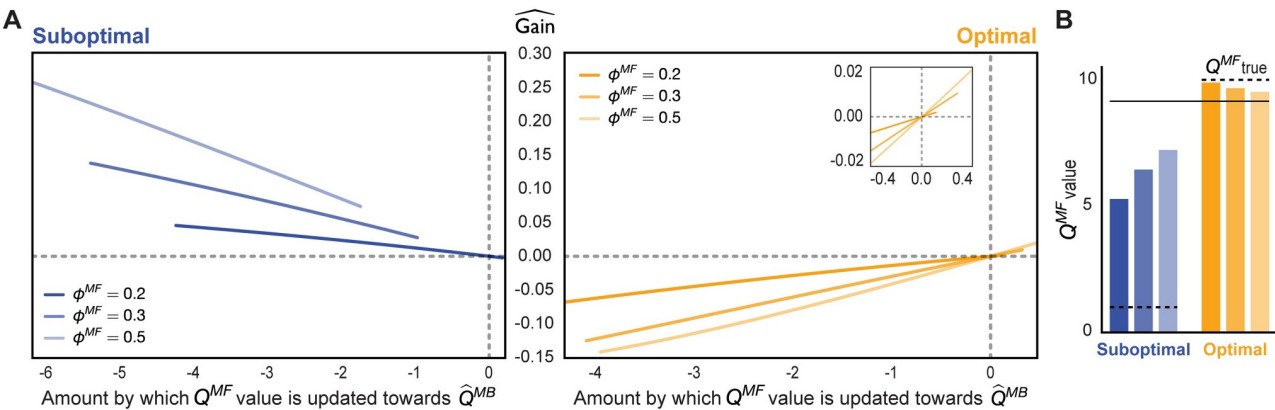

**Fig 4. How MF forgetting influences gain estimation.** (A) Estimated gain as a function of the difference between the agent's current MF $Q$-value and the model-estimated MB $Q$-value, $\widehat{Q}^{MB} - Q^{MF}$, for varying degrees of MF forgetting, $\phi^{MF}$. The dashed grey lines show the x- and y-intercepts. Note that the estimated gain is negative whenever the model-generated $\widehat{Q}^{MB}$ estimates are worse than the current MF $Q$-values. (B) Current MF $Q$-values for the optimal and sub-optimal actions with varying MF forgetting rate, coloured in the same way as above. The horizontal solid black bar is the average reward experienced so far, towards which MF values tend. The true $Q$-value for each action is shown in dashed black.

are better than their current MF $Q$-values. Forgetting in the model of the effects of actions (i.e., transition probabilities) will tend to homogenize the expected values of the actions—and this exerts a particular toll on the actions that are in fact the best. Conversely, forgetting of the effects of sub-optimal actions if anything makes them more prone to be replayed. This is because any sub-optimal action considered for replay will have positive estimated gain as long as MB $Q$-value for that action (as estimated by the agent's state-transition model) is less than its current MF $Q$-value. Due to MF forgetting, the agent's MF $Q$-values for sub-optimal actions rise above the actual average reward of the environment, and therefore, even if the state-transition model is uniform—which is the limit of complete forgetting of the transition matrix, the agent is still able to generate MB $Q$-values for sub-optimal actions that are less than their current MF $Q$-values, and use those in replay to improve the MF policy.

To examine these, we quantified uncertainty in the agent's state-transition model, for every state $s$ and action $m$ considered for replay, using the standard Shannon entropy [50]:

$$\mathbb{H}(s, m) = \mathop{\mathbb{E}}_{s' \sim T(s'|m,s)} \left[ -\log_2 T(s' \mid m, s) \right] \qquad (1)$$

over the potential states to which the agent could transition. Similarly, the agent's uncertainty about 2-move transitions considered for replay was computed as the joint entropy of the two transitions (Eq 23). For any state $s$ and action $m$ we will henceforth refer to $\mathbb{H}(s, m)$ as the action entropy (importantly, it is not the overall model entropy since the agent can be more or less uncertain about particular transitions). If an action that is considered for replay has high action entropy, the estimated MB $Q$-value of that action is corrupted by the possibility of transitioning into multiple states (Fig 5A, left); in fact, maximal action entropy corresponds to a uniform policy. For an action with low action entropy, on the other hand, the agent is able to estimate the MB $Q$-value of that action more faithfully (Fig 5A, right).

Thus, we examined how the estimated gain of objectively optimal and sub-optimal actions is determined by action entropy (Fig 5B). Indeed, we found that the estimated gain for optimal actions was positive only at low action entropy values (Fig 5B, right; see also S1 Fig), hence confirming that uncertainty in the agent's state-transition model significantly limited its ability to benefit from the replay of optimal actions. By contrast, the estimated gain for sub-optimal

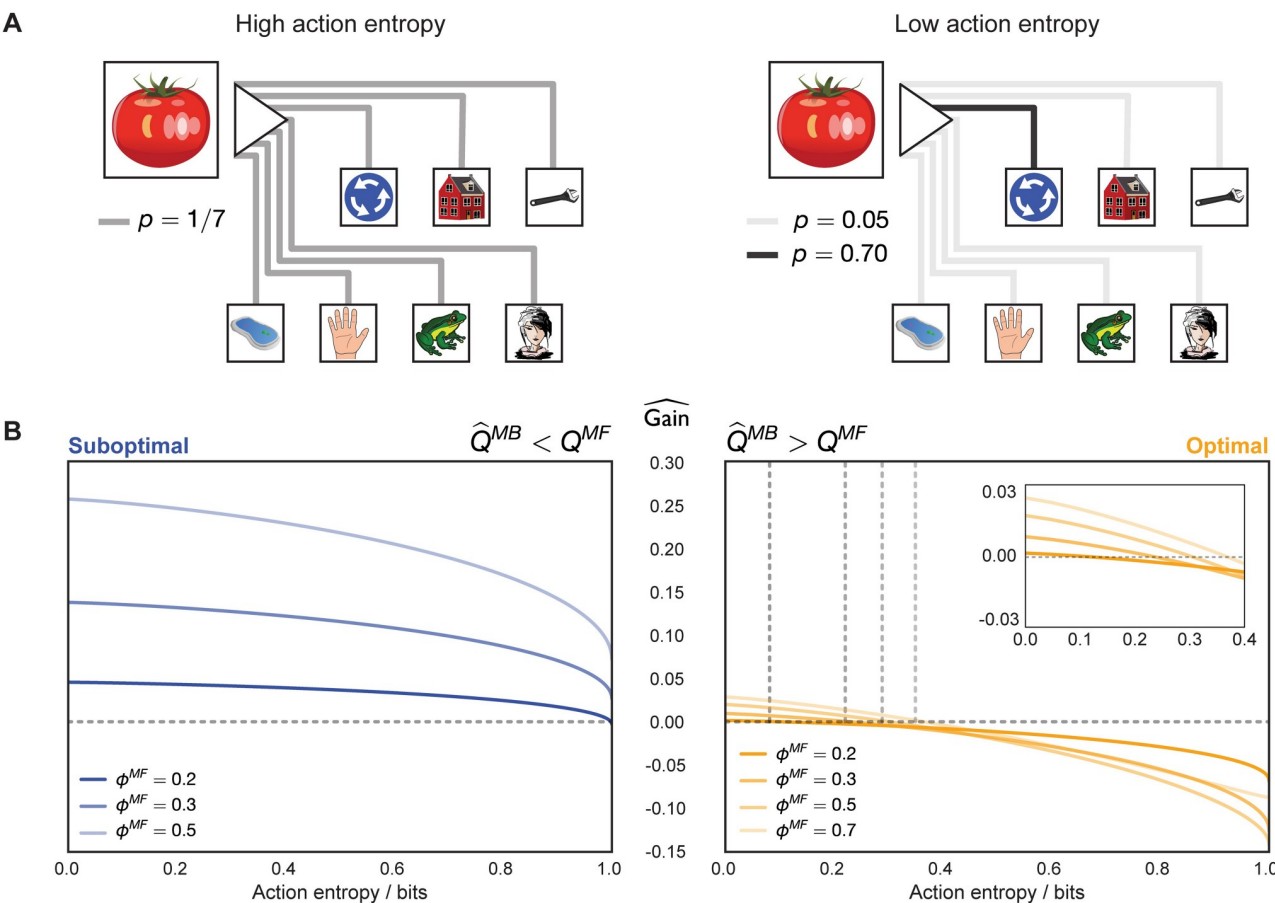

**Fig 5. Action entropy limits estimated gain.** (A) Left: under high action entropy, the distribution over the potential states to which the agent can transition given the current state and a chosen action is close to uniform. Right: under low action entropy, the agent is more certain about the state to which a chosen action will transition it. (B) Left: for an objectively sub-optimal action, the gain is positive throughout most action entropy values. Right: for an objectively optimal action, the gain becomes positive only when the state-transition model is sufficiently accurate. With heavier MF forgetting (higher $\phi^{MF}$), however, the intercept shifts such that the agent is able to benefit from a less accurate model (grey dashed lines show the x- and y-intercepts). The inset magnifies the estimated gain for the optimal action. Moreover, note how the magnitude of the estimated gain for an objectively optimal action is lower than that of a sub-optimal one, which is additionally influenced by the asymmetry of MF forgetting and on-line learning.

actions was positive for a wider range of action entropy values, compared to that for optimal actions (Fig 5B, left; see also S1 Fig).

Of course, optimal actions are performed more frequently, thus occasioning more learning. This implies that the state-transition model will be more accurate for the effects of those actions than for sub-optimal actions, which will partly counteract the effect evident in Fig 5. On the other hand, as discussed before, the estimated gain for optimal actions will be, in general, lower, precisely because of on-line learning as a result of the agent frequenting such actions.

An additional relevant observation is that action entropy (due to MB forgetting), together with MF forgetting, determines the balance between MF and MB strategies to which the agent apparently resorts. This can be seen from the increasing x-intercept as a function of the strength of MF forgetting for optimal actions (vertical dashed lines in Fig 5B, right). In the case of weak MF forgetting, the agent can only benefit from the replay of optimal actions inasmuch as the state-transition model is sufficiently accurate—which requires action entropy values to be low. As MF forgetting becomes more severe, the agent can think itself to benefit from replay

extracted from a worse state-transition model. Thus, we identify a range of parameter regimes which can lead agents to find replay subjectively beneficial and, therefore, allocate more influence to the MB system.

## Fitting actual subjects

We fit the free parameters of our model to the individual subject choices from the study of Eldar *et al.* (2020) [31], striving to keep as close as possible to the experimental conditions, for instance by treating the algorithm's adaptations to the image-reward association changes and the spatial re-arrangement in the same way as in the original study (see Methods for details). First, we examined whether our model correctly captured the varying degree of decision flexibility that was observed across subjects. Indeed, we found that the simulated IF values, as predicted by our agent with subject-tailored parameters, correlated significantly with the behavioural IF values (S5A Fig). Moreover, our agent predicted that some subjects would be engaging in potentially measurable replay (an average of more than 0.3 replays per trial, $n = 20$), and hence use MB knowledge when instructing their MF policies to make decisions. Indeed, we found that our agent predicted those subjects to be significantly less ignorant about the transition probabilities at the end of the training trials (2-sample 2-tailed t-test, $t = -2.56$, $p = 0.014$; S8 Fig), thus indicating the accumulation of MB knowledge. Further, those same subjects had significantly higher simulated IF values relative to the subjects for whom the agent did not predict sufficient replay—which is in line with the observation of Eldar *et al.* (2020) [31] that subjects with higher IF had higher MEG sequenceness following 'surprising' (measured by individually-fit state prediction errors) trial outcomes.

We then examined the replay patterns of those subjects, which we refer to as model-informed (MI) subjects, when modelled by our agent. There is an important technical difficulty in doing this exactly: our agent was modelled in an on-policy manner—i.e., making choices and performing replays based on its subjective gain, which, because of stochasticity, might not emulate those of the subjects, even if our model exactly captured the mechanisms governing choice and replay in the subjects. However, we can still hope for general illumination from the agent's behaviour.

In Fig 6A, we show an example move in a 1-move trial in the final block of the task which was predicted by our agent with parameters fit to an MI subject. This example is useful for demonstrating how the agent's forgetful state of knowledge (as discussed in the previous section) led it to prioritise the replay of certain experiences, and whether those choices were objectively optimal.

In this particular example, the agent chose a sub-optimal move. The agent's state of knowledge of 1-move transitions before and after executing the same move is shown in Fig 6B and 6C. Note how the agent is less ignorant, on average, about the outcomes of 1-move optimal actions, as opposed to sub-optimal ones (Fig 6B). The agent chose a sub-optimal move because the MF $Q$-value for that move had been forgotten to an extent that the agent's subjective knowledge incorrectly indicated it to be optimal (Fig 6C). After learning online that the chosen move was worse than predicted (Fig 6C, compare green and blue bars), the agent replayed that action at the end of the trial to incorporate the estimated MB $Q$-value for that action, as generated by the agent's state-transition model (Fig 6C, pink bar), into its MF policy (Fig 6D). In this case, the MB knowledge that the agent decided to incorporate into its MF policy was more accurate as regards the true $Q$-values; such replay, therefore, made the agent less likely to re-choose the same sub-optimal move in the following trials. Therefore, in this example, our agent predicted that the subject engaged in significant replay of the recent sub-optimal single move experience.

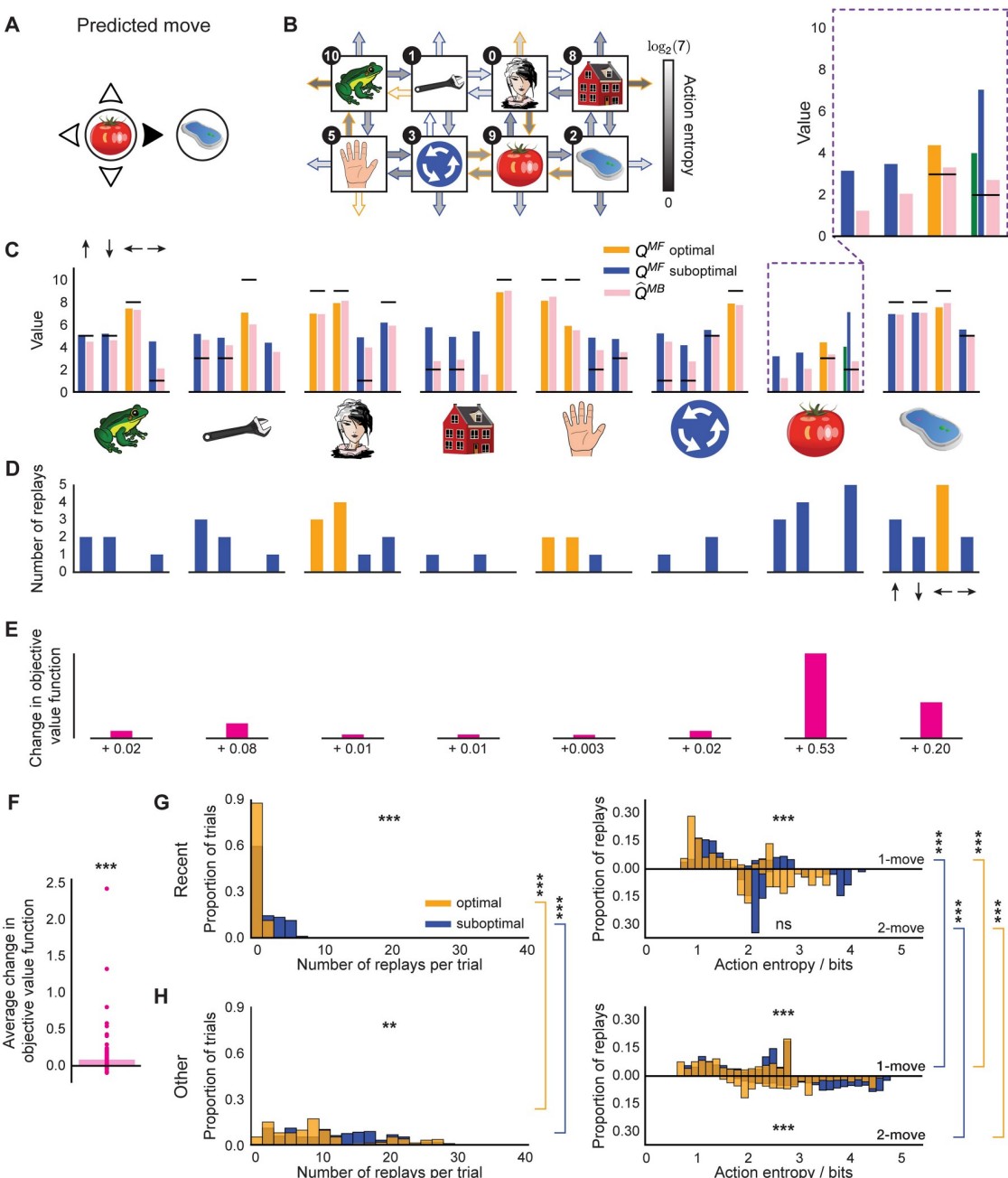

**Fig 6. Epistemology of replay.** (A) An example move predicted by our agent with subject-specific parameters. (B) State-transition model of the agent after executing the move in (A) and the associated action entropy values. Objectively optimal actions are shown as arrows with orange outlines; sub-optimal—with blue outlines. (C) State of MF and MB knowledge of the agent. The arrows above the leftmost bar plot indicate the directions of the corresponding actions in each plot. The horizontal black lines represent the true reward obtainable for each action. The agent's knowledge at the state where the trial began is highlighted in a purple dashed box and is additionally magnified above. The blue bar for the MF $Q$-value that corresponds to the predicted move in (A) shows what the agent knew before executing the move, and the neighbouring green bar—what the agent has learnt on-line after executing the move (note that the agent always learnt on-line towards the true reward). (D) Replay choices of the agent. (E) Changes in the objective value function (relative to the true obtainable reward) of each state as a result of the replay in (D), not drawn to scale. (F) Same as in (D) but across the entire experiment and averaged over all states. (G-H) Average replay statistics over the entire experiment. (G) Just-experienced transitions; (H) Other transitions. First column: proportion of sub-optimal and optimal trials in which objectively sub-optimal or optimal action(s) were replayed. Second column: proportion of action entropy values at which the replays were executed. Upper and lower y-axes show the action entropy distribution for 1-move and 2-move trials respectively. $^{**}$ $p < 0.01$, $^{***}$ $p < 0.001$, ns: not significant.

As just discussed, in addition to replaying the just-experienced transition, the agent also engaged in the replay of other transitions. Indeed, the agent was not restricted in which transitions to replay—it chose to replay actions based solely on whether the magnitude of the estimated gain of each possible action is greater than a subject-specific gain threshold (see Methods for details). We were, therefore, able to see a much richer picture of the replay of all allowable transitions, in addition to the just-experienced ones. In the given example, it is easy to see how the agent estimated its replay choices to be gainful: because of the relatively strong MF forgetting and a sufficiently accurate state-transition model. To demonstrate a slightly different parameter regime, we also additionally show an example move predicted by our agent for another MI subject (S4 Fig). In that case, the agent's MF and MB policies were more accurate; however, our parameter estimates indicated that the subject's gain threshold for initiating replay was set lower, and hence the agent still engaged in replay, even though the gain it estimated was apparently minute.

To quantify the objective benefit of replay for the subject shown in Fig 6, we examined how the agent's objective value function (relative to the true obtainable reward) at each state changed due to the replay at the end of this trial, which is a direct measure of the change in the expected reward the agent can obtain. We found that for each state where the replay occurred, the objective value function of that state increased (Fig 6E). To see whether such value function improvements held across the entire session, we examined the average trial-wise statistics for this subject (Fig 6F). We found that, on average, the replay after each trial (both 1-move and 2-move) improved the agent's objective value function by 0.08 reward points (1-sample 2-tailed t-test, $t = 6.34$, $p \ll 0.0001$).

We next looked at the average trial-wise replay statistics of the subject as predicted by our agent (Fig 6G and 6H). If we consider solely the just-experienced transitions (Fig 6G), we found that there were significantly more replays of sub-optimal actions per trial (Wilcoxon rank-sum test, $W = 4.71$, $p = 2.51 \cdot 10^{-6}$). Moreover, the replay of sub-optimal single actions was at significantly higher action entropy values (Wilcoxon rank-sum test, $W = 5.58$, $p \ll 0.0001$), which is what one would expect given that the transitions that led to optimal actions were experienced more frequently. Since some optimal 1-move actions corresponded to sub-optimal first moves in 2-move trials, the agent received an additional on-line training about the latter transitions, and we therefore found no significant difference in action entropy values at which coupled optimal and sub-optimal actions were replayed (Wilcoxon rank-sum test, $W = -0.11$, $p = 0.91$).

In addition to the just-experienced transitions, we also separately analysed the replay of all other transitions (Fig 6H); this revealed a much broader picture, but we observed the same tendency for sub-optimal actions to be replayed more (Wilcoxon rank-sum test, $W = 2.88$, $p = 0.004$). In this case, we found that both 1-move (Wilcoxon rank-sum test, $W = 8.31$, $p \ll 0.0001$) and 2-move (Wilcoxon rank-sum test, $W = 24.7$, $p \ll 0.0001$) sub-optimal actions were replayed at higher action entropy values than the corresponding optimal actions. We also compared the replays of recent and 'other' transitions and their entropy values. We found that both optimal other (Wilcoxon rank-sum test, $W = 19.5$, $p \ll 0.0001$) and sub-optimal other (Wilcoxon rank-sum test, $W = 12.6$, $p \ll 0.0001$) transitions were replayed more than the recent ones. All 1-move and 2-move optimal and sub-optimal other transitions were replayed at significantly higher action entropy values than the corresponding recent transitions (1-move optimal recent vs 1-move optimal other, Wilcoxon rank-sum test, $W = 9.25$, $p = \ll 0.0001$; 1-move sub-optimal recent vs 1-move sub-optimal other, Wilcoxon rank-sum test, $W = 14.7$, $p \ll 0.0001$, 2-move optimal recent vs 2-move optimal other, Wilcoxon rank-sum test, $W = 3.36$, $p = 7.77 \cdot 10^{-4}$; 2-move sub-optimal recent vs 2-move sub-optimal other, Wilcoxon rank-sum test, $W = 8.47$, $p \ll 0.0001$). The latter observation could potentially

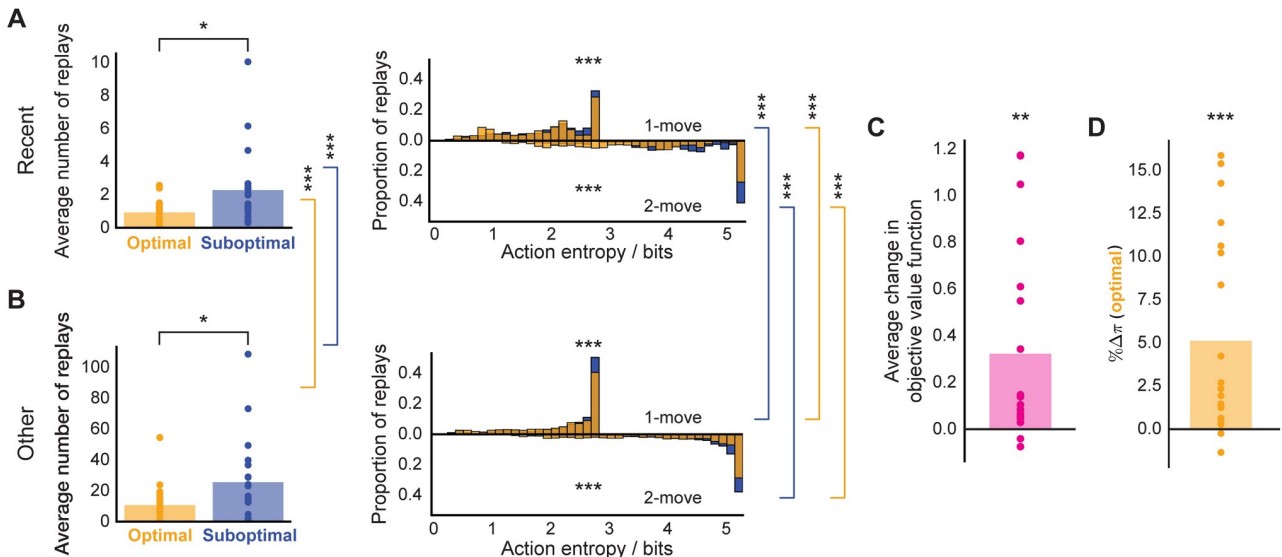

**Fig 7. Overall on-task replay statistics across MI subjects.** (A) Left: average number of replays of just-experienced optimal and sub-optimal actions; right: proportion of action entropy values at which just-experienced optimal and sub-optimal actions were replayed. (B) Same as above but for other transitions. (C) Average change in objective value function due to replay. (D) Average change in the probability of selecting an (objectively) optimal action due to replay. * $p < 0.05$, ** $p < 0.01$, *** $p < 0.001$.

explain why these 'distal' replays of other transitions were not detectable in the study of Eldar *et al.* (2020) [31], since the content of highly entropic on-task replay events may have been improbable to identify with classifiers trained on data obtained during the pre-task stimulus exposure (while subjects were contemplating the images with certitude).

Overall, we found that the pattern of the replay choices selected by our agent with parameters fit to the data for the subject shown in Fig 6 was consistent with the observations reported by Eldar *et al.* (2020) [31]. Furthermore, this consistency was true across all subjects with significant replay (Fig 7). Each MI subject, on average, preferentially replayed sub-optimal actions (just-experienced transitions, Wilcoxon rank-sum test, $W = 2.48$, $p = 0.012$; other transitions, Wilcoxon rank-sum test, $W = 2.33$, $p = 0.020$). As for the subject in Fig 6, we assessed whether the action entropy associated with the just-experienced transition (be it optimal or sub-optimal) when it was replayed was lower than the action entropy associated with other optimal and sub-optimal actions when they were replayed. Overall, we found the same trend across all MI subjects but for all combinations: i.e., 1-move transitions (just-experienced 1-move optimal vs other 1-move optimal, Wilcoxon rank-sum test, $W = 11.3$, $p \ll 0.0001$; just-experienced 1-move sub-optimal vs other 1-move sub-optimal, Wilcoxon rank-sum test, $W = 11.8$, $p \ll 0.0001$) and 2-move transitions (just-experienced 2-move optimal vs other 2-move optimal, Wilcoxon rank-sum test, $W = 14.2$, $p \ll 0.0001$; just-experienced 2-move sub-optimal vs other 2-move sub-optimal, Wilcoxon rank-sum test, $W = 24.7$, $p \ll 0.0001$). Furthermore, our agent predicted that, in each trial, MI human subjects increased their (objective) average value function by 0.323 reward points (1-sample 2-tailed t-test, $t = 3.47$, $p = 0.003$, Fig 7C) as a result of replay. As a final measure of the replay benefit, we quantified the average trial-wise increase in the probability of choosing an optimal action as a result of replay (Fig 7D). This was done to provide a direct measure of the effect of replay on the subjects' policy, as opposed to the proxy reported by Eldar *et al.* (2020) [31] in terms of the visitation frequency (shown in Fig 1B). On average, our modelling showed that replay increased the probability of choosing an optimal

action in MI subjects by 5.11% (1-sample 2-tailed t-test, $t = 3.93$, $p = 0.0008$), which is very similar to the numbers reported by Eldar *et al.* (2020) [31].

Clearly, our modelling predicted quite a diverse extent to which MI subjects objectively benefited (or even hurt themselves) by replay (Fig 7C and 7D). This is because the subjects had to rely on their forgetful and imperfect state-transition models to estimate MB *Q*-values for each action, and as a result of MB forgetting, some subjects occasionally mis-estimated some sub-optimal actions to be optimal—and thus their average objective replay benefit was more modest (or even negative in the most extreme cases). Indeed, upon closer examination of our best-fitting parameter estimates for each MI subject, we noted that subjects who, on average, hurt themselves by replay ($n = 2$) had very high MB forgetting rates (S3D Fig), such that their forgetting-degraded models further exacerbated their knowledge of the degree to which each action was rewarding (the example subject shown in S5 Fig has this characteristic). Moreover, we found a significant anticorrelation of MB forgetting parameter and IF in MI subjects (multiple regression, $\beta = -0.37$, $t = -2.73$, $p = 0.016$; S3B Fig), which suggests that MI subjects with a more accurate state-transition model achieved higher decision flexibility by preferentially engaging in pessimized replay following each trial outcome.

We did not find any significant linear relationship between the gain threshold and objective replay benefit (multiple regression, $\beta = 0.24$, $t = 1.86$, $p = 0.083$; S3F Fig). This suggests that the effect is either non-linear or highly dependent on the current state of knowledge of the agent (due to the on-policy nature of replay in our algorithm), and thus higher MB forgetting rate could not have been ameliorated by simply having a stricter gain threshold. Moreover, we note that extensive training of MF values by a forgetful MB policy (as a result of setting the gain threshold too low) can be dangerous. For instance, our modelling predicted the example subject in S5 Fig to be close to complete ignorance about the transition structure of the task (S5B Fig), yet the subject, as our agent predicted, still chose to engage in replay which significantly hurt his performance (S5F Fig).

The apparent exuberance of distal replays that we discovered suggested that our model fit predicted the subjects to be notably more forgetful than found by Eldar *et al.* (2020) [31]. We thus compared our best-fitting MB and MF forgetting parameter estimates to those reported by Eldar *et al.* (2020) [31] for the best-fitting model. We found that our model predicted most subjects to be subject to far less MB forgetting (S2B Fig). On the other hand, our MF forgetting parameter estimates implied that the subjects remembered their MF *Q*-values significantly less well (S2A Fig). We (partially) attribute these differences to optimal replay: in Eldar *et al.* (2020) [31] the state-transition model explicitly affected every choice, whereas in our model, the MF (decision) policy was only informed by MB quantities so long as this influence was estimated to be gainful. Thus, the MF policy could be more forgetful, and yet be corrected by subjectively optimal MB information; this is also supported by the significant correlation of MF forgetting and objective replay benefit (multiple regression, $\beta = 0.81$, $t = 9.33$, $p \ll 0.0001$; S3F Fig). Moreover, due to substantially larger MF forgetting predicted by our modelling, we predicted that the subjects would have been more 'surprised' by how rewarding each outcome was. The combination of this surprise with the (albeit reduced) suprise about the transition probabilities, arising from the (lower) MB forgetting, resulted in surplus distal replays.

## Discussion

In summary, we studied the consequences of forgetting in a DYNA-like agent [39] with optimised replay [41]. The agent uses on-line experiences to train both its model-free policy (by learning about the rewards associated with each action) and its model-based system (by (re) learning the transition probabilities). It uses off-line experience to allow MF values to be

trained by the MB system in a supervised manner. Behaviour is ultimately controlled exclusively by the MF system. The progressive inaccuracy of MF values (as a result of forgetting) can be ameliorated by MB replay, but only if the MB system has itself not become too inaccurate.

In particular, we showed that the structure of forgetting could favour the replay of sub-optimal rather than optimal actions. This arose through the interaction of several factors. One is that MF values relax to the mean experienced reward (meaning that sub-optimal actions come to look better, and optimal actions look worse than they actually are)–this can lead to the progressive choice of sub-optimal actions, and thus gain in suppressing them. A second is that optimal actions generally enjoy greater updating from actual on-line experience, since they are chosen more frequently. A third is that MB values relax towards the mean of all the rewards in the environment (because the transition probabilities relax towards uniformity), which is pessimistic relative to the experienced rewards. This makes the model relatively worse at elevating optimal actions in the MF system. We showed that the Shannon entropy of the transition distribution was a useful indicator of the status of forgetting.

We found that replay can both help and hurt, from an objective perspective—with the latter occurring when MB forgetting is too severe. This shows that replay can be dangerous when subjects lack meta-cognitive monitoring insight to be able to question the veracity of their model and thus the benefit of using it (via subjective gain estimates). This result has implications for sub-optimal replay as a potential computational marker of mental dysfunction [51, 52].

We studied these phenomena in the task of Eldar *et al.* (2020) [31], where it was initially identified. Forgetting had already been identified by Eldar *et al.* (2020) [31] (in fact, a number of previous studies had also noted the beneficial effect of value forgetting on model fits to behavioural data [53–55]); however, it played a more significant role in our model, because we closed the circle of having MB replay affect MF values and thus behaviour. By not doing this, forgetting may have been artificially downplayed in the original report (S2A and S2B Fig).

Additionally, we note that our forgetting mechanism first arose in the literature on computational and behavioural neuroscience; average reward rate, for instance, has been found to be an important facet of behaviour—being used as the opportunity cost of time in accounts of controlled vigour [56]. Unfortunately, the task is not very well adapted to address broader psychological questions of forgetting [57], since opportunities for decay and prospective and retrospective interference all abound in the trial-type sequences employed. Furthermore, longer-term consolidation was not a focus.

We fit the free parameters of our agent to the behavioural data from individual subjects, correctly capturing their behavioural flexibility (IF) (S3A and S3B Fig), and providing a mechanistic explanation for the replay choice preferences of those employing a hybrid MB/MF strategy. Our fits suggested that these MI subjects underwent significantly more MF forgetting than those whose behavior was more purely MF (since the latter could only rely on the MF system, see S3C Fig).

Our study has some limitations. First, we shared Mattar and Daw (2018)'s [41] assumption that the subjects could compute gain correctly in their resulting incorrect models of the task. How gain might actually be estimated and whether there might be systematic errors in the calculations involved are not clear. However, we note that spiking neural networks have recently proven particularly promising for the study of biologically plausible mechanisms that support inferential planning [58, 59], and they could therefore provide insights into the underlying computations that prioritise the replay of certain experiences.

Second, our model did not account for the need term [41] that was theorised to be another crucial factor for the replay choices. This was unnecessary for the current task. Need is closely

related to the successor representation (SR) of sequential transitions [49] inasmuch as it predicts how often one expects each state to be visited given the current policy. Need was shown to mostly influence forward replay at the very outset of a trial as part of planning, something that is only just starting to be detectable using MEG [60].

The SR also has other uses in the context of planning, and is an important intermediate representation for various RL tasks. Furthermore, it has a similar computational structure to MF $Q$-values—in fact, it can be acquired through a form of MF Q learning with a particular collection of reward functions [49]. Thus, it is an intriguing possibility, suggested, for instance, in the SR-DYNA algorithm of Russek *et al.* (2017) [61] or by the provocative experiment of Carey *et al.* (2019) [62] that there may be forms of MB replay that are directed at maintaining the integrity and fidelity of the SR in the face of forgetting and environmental change. Indeed, grid cells in the superficial layers of entorhinal cortex were shown to engage in replay independently of the hippocampus, and thus could be a potential candidate for SR-only replay [63, 64]. More generally, other forms of off-line consolidation could be involved in tuning and nurturing cognitive maps of the environment, leading to spatially-coherent replay [65].

Third, Eldar *et al.* (2020) [31] also looked at replay during the 2-minute rest period that preceded each block, finding an anticorrelation between IF and replay. The most interesting such periods preceded blocks 3 and 5, after the changes to the reward and transition models. Before block 3, MF subjects replayed transitions that they had experienced in block 2 and preplayed transitions they were about to choose in block 3; MI subjects, by contrast, showed no such bias. Although we did show that even if we set the MF $Q$-values to zero before allowing the agent to engage in replay prior to starting blocks 3 and 5 (S7 Fig), we still recover our main conclusions, the fact that we do not know what sort of replay might be happening during retraining meant that we left modelling this rest-based replay to future work.

Finally, for consistency, we borrowed some of the assumptions made in the study of Eldar *et al.* (2020) [31]. In particular, we did not not consider other sources of uncertainty which could have arisen from, for instance, forgetting of image-reward associations (since subjects were extensively pre-trained and, furthermore, Eldar *et al.* (2020) [31] validated an explicit recall of those associations in all participants upon completion of the task). Nevertheless, we note that such a mechanism could have impacted the observed pessimized pattern of replay. Pessimistic replay would prevail so long as the stimulus-specific rewards are forgotten towards a value which is less than the average experienced reward (with our form of MF forgetting).

To summarize, in this work, we showed both in simulations and by fitting human data in a simple planning task that pessimized replay can have distinctly beneficial effects. We also showed the delicacy of the balance of and interaction between MB and MF systems when they are forgetful—something that will be of particular importance in more sophisticated and non-deterministic environments that involve partial observability and large state spaces [66, 67].

## Methods

### Task design

We simulated the agent in the same task environment and trial and reward structure as Eldar *et al.* (2020) [31]. The task space contained 8 states arranged in a $2 \times 4$ torus, with each state associated with a certain number of reward points. From each state, 4 actions were available to the agent—up, down, left, and right. If the state the agent transitioned to was revealed (trials with feedback), the agent was awarded reward points associated with the state to which it transitioned.

The agent started each trial in the same location as the corresponding human subject, which was originally determined in a pseudo-random fashion. The agent then chose among

the available moves. Based on the chosen move, the agent transitioned to a new state and received the reward associated with that state. In 1-move trials, this reward signified the end of the trial, whereas in 2-move trials the agent then had to choose a second action either with or without feedback as to which state the first action had led. In addition, transitions back to the start state in 2-move trials were not allowed. Importantly, the design of the task ensured that optimal first moves in 2-move trials were usually different from optimal moves in 1-move trials.

As for each of the human subjects, the simulation consisted of 5 task blocks that were preceded by 6 training blocks. Each training block consisted of twelve 1-move trials from 1 of 2 possible locations. The final training block contained 48 trials where the agent started in any of the 8 possible locations. In the main task, each block comprised 3 epochs, each containing six 1-move trials followed by twelve 2-move trials, therefore giving in total 54 trials per block. Every 6 consecutive trials the agent started in a different location except for the first 24 2-move trials of the first block, in which Eldar *et al.* (2020) [31] repeated each starting location for two consecutive trials to promote learning of coupled moves. eginning with block 2 the agent did not receive any reward or transition feedback for the first 12 trials of each block. After block 2 the agent was instructed about changes in the reward associated with each state. Similarly, before starting block 5 the agent was informed about a rearrangement of the states in the torus. Eldar *et al.* (2020) [31] specified this such that the optimal first move in 15 out of consecutive 16 trials became different.

## DYNA-like agent

**Free parameters.** The potential free parameters of our model are listed in Table 1.

**Choices.** The agent has the same model-free mechanism for choice as designed by Eldar *et al.* (2020) [31]. However, unlike Eldar *et al.* (2020) [31], choices are determined exclusively by the MF system; the only role the MB system plays is via replay, updating the MF values.

The model-free system involves two sets of state-action or $Q^{MF}$ values [48]: $Q^{MF1}$ for single moves (the only move in 1-move trials, and both first and second moves in 2-move trials), and $Q^{MF2}$ for coupled moves (2-move sequences) in 2-move trials.

**Table 1. Free parameters of the algorithm.**

| | |
|---:|:---|
| $\eta^{MF1}$ | learning rate in 1-move trials and second moves in 2-move trials |
| $\eta^{MF2}$ | learning rate in first moves in 2-move trials |
| $\eta_r^{MF1}$ | learning rate for replay in 1-move trials and second moves in 2-move trials |
| $\eta_r^{MF2}$ | learning rate for replay in first moves in 2-move trials |
| $\beta_1^{MF1}$ | inverse temperature in 1-move trials |
| $\beta_2^{MF1}$ | inverse temperature for second moves in 2-move trials |
| $\beta_2^{MF2}$ | inverse temperature for first moves in 2-move trials |
| $\theta$ | initialisation mean for $Q^{MF}$ values |
| $\gamma_m$ | fixed bias for each action, subject to $\Sigma_m \, \gamma_a = 0$ |
| $\eta^{MB}$ | state-transition model learning rate |
| $\rho$ | fraction of the state-transition model learning rate that is used for updating opposite transitions |
| $\phi^{MB}$ | state-transition model forgetting |
| $\phi'^{MB}$ | state-transition model forgetting upon spatial re-arrangement |
| $\omega$ | state-transition model re-arrangement success |
| $\phi^{MF}$ | $Q^{MF}$ values forgetting |
| $\phi'^{MF}$ | $Q^{MF}$ values forgetting upon spatial re-arrangement |
| $\xi$ | gain threshold for initiating replay |

In 1-move trials, starting from state $s_{t,1}$, the agent chooses actions according to the softmax policy:

$$\pi(m_{t,1} = m \mid s_{t,1}) \propto e^{\gamma_m + \beta_1^{MF1} Q_t^{MF1}(s_{t,1}, m)} \tag{2}$$

where $m \in \{up, down, left, right\}$. In 2-move trials, the first move is chosen based on the combination of both sorts of $Q^{MF}$ values:

$$\pi(m_{t,1} = m \mid s_{t,1}) \propto e^{\gamma_m + \beta_2^{MF1} Q_t^{MF1}(s_{t,1}, m) + \beta_2^{MF2} Q_t^{MF2}(s_{t,1}, m)} \tag{3}$$

where the individual $Q^{MF1}$ value is weighted by a different inverse temperature or strength parameter $\beta_2^{MF1}$, and the coupled move $Q_t^{MF2}(s_{t,1}, m)$ value comes from considering all possible second moves $m_{t,2}$, weighted by their probabilities:

$$Q_t^{MF2}(s_{t,1}, m) = \sum_{m'} \pi(m_{t,2} = m' \mid s_{t,1}, m_{t,1}) Q_t^{MF2}(s_{t,1}, m, m') \tag{4}$$

The two-argument $Q^{MF2}$ value used when choosing a first move therefore corresponds to a two-move MF Q value where the second move options are marginalised out with respect to the agent's policy.

When choosing a second move the agent takes into partial account (since $Q^{MF2}$ doesn't) the state $s_{t,2}$ to which it transitioned on the first move:

$$\pi(m_{t,2} = m \mid s_{t,1}, m_{t,1}, s_{t,2}) \propto e^{\gamma_m + \beta_2^{MF1} Q_t^{MF1}(s_{t,2}, m) + \beta_2^{MF2} Q_t^{MF2}(s_{t,1}, m_{t,1}, m)} \tag{5}$$

In trials without feedback, since the agent's $Q^{MF}$ values are indexed by state and the transition is not revealed, it is necessary in Eq 5 to average $Q^{MF1}$ over all the permitted $s_{t,2}$ (since returning to $s_{t,1}$ is disallowed):

$$\langle Q_t^{MF1}(s_{t,2}, m) \rangle = \frac{1}{7} \sum_s Q_t^{MF1}(s, m) \tag{6}$$

The decision is then made according to Eq 5.

**Model-free value learning.** The reward $R(s_{t,2})$ received for a given move $m_t$ and transition to state $s_{t,2}$ is used to update the agent's $Q^{MF1}$ value for that move:

$$Q_{t+1}^{MF1}(s_{t,1}, m_t) \leftarrow Q_t^{MF1}(s_{t,1}, m_t) + \eta^{MF1}[R(s_{t,2}) - Q_t^{MF1}(s_{t,1}, m_t)] \tag{7}$$

On 2-move trials, the same rule is applied for the second move, adjusting $Q_{t+1}^{MF1}(s_{t,2}, m_{t,2})$ according to the second reward $R(s_{t,3})$. Furthermore, on 2-move trials, $Q^{MF2}$ values are updated at the end of each trial based on the sum total reward obtained on that trial but with a different learning rate:

$$
\begin{aligned}
Q_{t+1}^{MF2}(s_{t,1}, m_{t,1}, m_{t,2}) \leftarrow \quad & Q_t^{MF2}(s_{t,1}, m_{t,1}, m_{t,2}) + \\
& \eta^{MF2}[R(s_{t,2}) + R(s_{t,3}) - Q_t^{MF2}(s_{t,1}, m_{t,1}, m_{t,2})]
\end{aligned} \tag{8}
$$

**Model-based learning.** Additionally, and also as in Eldar *et al.* (2020) [31], the agent's model-based system learns about the transitions associated with every move it experiences (i.e. the only transition in 1-move trials and both transitions in 2-move trials):

$$T_{t+1}(s_{t,1}, m_t, s_{t,2}) \leftarrow T_t(s_{t,1}, m_t, s_{t,2}) + \eta^{MB}[1 - T_t(s_{t,1}, m_t, s_{t,2})] \tag{9}$$

and, given that actions are reversible, learning also happens for the opposite transitions to a

degree that is controlled by parameter $\rho$:

$$T_{t+1}(s_{t,2}, \tilde{m}_t, s_{t,1}) \leftarrow T_t(s_{t,2}, \tilde{m}_t, s_{t,1}) + \rho\eta^{MB}[1 - T_t(s_{t,2}, \tilde{m}_t, s_{t,1})] \tag{10}$$

where $\tilde{m}_t$ is the transition opposite to $m_t$. Note that in trials with no feedback the agent does not receive any reward and the state it transitions to is uncued. Therefore, no learning occurs in such trials.

To ensure that the probabilities sum up to 1, the agent re-normalizes the state-transition model after every update and following the MB forgetting (see below) as:

$$\forall s \quad T_{t+1}(s_{t,1}, m_t, s) \leftarrow \frac{T_{t+1}(s_{t,1}, m_t, s)}{\sum_{s'} T_{t+1}(s_{t,1}, m_t, s')} \tag{11}$$

**Replay.** That our agent exploits replay is the critical difference from the model that Eldar *et al.* (2020) [31] used to characterize their subjects' decision processes.

Our algorithm makes use of its (imperfect) knowledge of the transition structure of the environment to perform additional learning in the inter-trial intervals by means of generative replay. Specifically, the agent utilises its state-transition model $T$ and reward function $R(s)$ to estimate model-based $\widehat{Q}^{MB}$ values for every possible action (that are allowable according to the model). These $\widehat{Q}^{MB}$ values are then assessed for the potential MF policy improvements (see below).

$\widehat{Q}^{MB}$ values for 1-move trials and second moves in 2-move trials are estimated as follows:

$$\widehat{Q}^{MB1}(s_1, m_1) = \sum_{s'} T(s_1, m_1, s')R(s') \tag{12}$$

Similarly, $\widehat{Q}^{MB}$ values for 2-move sequences are estimated as:

$$\widehat{Q}^{MB2}(s_1, m_1, m_2) = \sum_{s'} T(s_1, m_1, s')[R(s') + \sum_{s''} T(s', m_2, s'')R(s'')] \tag{13}$$

When summing over the potential outcomes for a second action in Eq 13, the agent additionally sets the probability of transitioning into the starting location $s_1$ to zero (since backtracking was not allowed) and normalises the transition probabilities according to Eq 11. Note that reward function $R(s)$ here is the true reward the agent would have received for transitioning into state $s$ since we assume that the subjects have learnt the image-reward associations perfectly well. The model-generated $\widehat{Q}^{MB}$ values therefore incorporate the agent's uncertainty about the transition structure of the environment. If the agent is certain which state a given action would take it to, $\widehat{Q}^{MB}$ value for that action would closely match the true reward function of that state. Otherwise, $\widehat{Q}^{MB}$ values for uncertain transitions are corrupted by the possibility of ending up in different states.

The agent then uses all the generated $\widehat{Q}^{MB}$ values to compute new hybrid $Q^{MF/MB}$ values. These hybrid values correspond to the values that would have resulted had the current $Q^{MF}$ values been updated towards the model-generated $\widehat{Q}^{MB}$ values using replay-specific learning

rates $\eta_r^{MF1}, \eta_r^{MF2}$:

$$Q^{MF1/MB1}(s_1, m_1) \leftarrow Q^{MF1}(s_1, m_1) + \eta_r^{MF1}[\widehat{Q}^{MB1}(s_1, m_1) - Q^{MF1}(s_1, m_1)] \tag{14}$$

$$\begin{aligned} Q^{MF2/MB2}(s_1, m_1, m_2) \leftarrow \quad & Q^{MF2}(s_1, m_1, m_2) + \\ & \eta_r^{MF2}[\widehat{Q}^{MB2}(s_1, m_1, m_2) - Q^{MF2}(s_1, m_1, m_2)] \end{aligned} \tag{15}$$

We note that the 2-move updates specified in Eq 15 are a form of supervised learning, and so differ from the RL/DYNA-based episodic replay suggested by Mattar and Daw (2018) [41]. As mentioned above, we chose this way of updating MF $Q$-values for coupled moves in replay to keep the algorithmic details as close to Eldar *et al.* (2020) [31] as possible. In principle, we could have also operationalised our 2-move replay in a DYNA fashion.

To assess whether any of the above updates improve the agent's MF policy, the agent computes the expected value of every potential update [41]:

$$\begin{aligned} \widehat{\text{Gain}}(s_1, m_1) = \quad & \sum_m Q^{MF1/MB1}(s_1, m)\pi_{new}(m \mid s_1) - \\ & \sum_m Q^{MF1/MB1}(s_1, m)\pi_{old}(m \mid s_1) \end{aligned} \tag{16}$$

Analogously, for a sequence of two moves $\{m_1, m_2\}$:

$$\begin{aligned} \widehat{\text{Gain}}(s_1, m_1, m_2) = \quad & \sum_{\{m_1, m_2\}} Q^{MF2/MB2}(s_1, m_1, m_2)\pi_{new}(\{m_1, m_2\} \mid s_1) - \\ & \sum_{\{m_1, m_2\}} Q^{MF2/MB2}(s_1, m_1, m_2)\pi_{old}(\{m_1, m_2\} \mid s_1) \end{aligned} \tag{17}$$

where the policy $\pi$ was assumed to be unbiased and computed as in Eq 2; that is, the agent directly estimated the corresponding probabilities for each sequence of 2 actions in 2-move trials. Both of these expressions for estimating $\widehat{\text{Gain}}$ use the full *new* model-free policy that would be implied by the update. Thus, as also in Mattar and Daw (2018) [41], the gain (Eqs 16 and 17) does not assess a psychologically-credible gain, since the new policy is only available *after* the replays are executed. Moreover, we emphasise that this same gain is the agent's estimate, for the true gain is only accessible to an agent with perfect knowledge of the transition structure of the environment (which is infeasible in the presence of substantial forgetting).

Finally, the expected value of each backup (EVB), or replay, is computed as $\frac{1}{8} \cdot \widehat{\text{Gain}}(\cdot)$ (since the agent starts each trial in a pseudorandom location, we assumed need to be uniform). Exactly as in Mattar and Daw (2018) [41], the priority of the potential updates is determined by the EVB value—if the greatest EVB value exceeds gain threshold $\xi$ (for simplicity and due to the assumption of uniform need, we refer to $\xi$ as gain threshold, rather than EVB threshold), then the agent executes the replay associated with that EVB towards the model-generated $\widehat{Q}^{MB}$ value according to Eqs 14 or 15 (depending whether it is a single move or a two-move sequence), thus incorporating its MB knowledge into the current MF policy. Note that since this changes the agent's MF policy and the generation of $\widehat{Q}^{MB2}$ is policy-dependent, the latter are re-generated following every executed backup. The replay proceeds until no potential updates have the EVB value greater than $\xi$.

**Forgetting.** The agent is assumed to forget both about the $Q^{MF}$ values and the state-transition model $T$ in trials where feedback is provided. Thus, after every update and following

replay the agent forgets according to:

$$Q_{t+1}^{MF} \quad \leftarrow (1 - \phi^{MF})Q_{t+1}^{MF} + \phi^{MF}\bar{r}_{t+1} \tag{18}$$

$$T_{t+1} \quad \leftarrow (1 - \phi^{MB})T_{t+1} + \phi^{MB}1/7 \tag{19}$$

Note that we parameterize the above equations in terms of forgetting parameters $\phi$. Eldar *et al.* (2020) [31] instead used $\tau$ as remembrance, or value retention, parameters. Therefore, our forgetting parameters $\phi$ are equivalent to Eldar *et al.* (2020) [31]'s $1 - \tau$.

The state-transition model therefore decays towards the uniform distribution over the potential states the agent can transition to given any pair of state and action. $Q^{MF}$ values are forgotten towards the average reward experienced since the beginning of the task, $\bar{r}_{t+1}$. For $Q^{MF1}$ values, it is the average reward obtained in single moves:

$$\bar{r}_{t+1} = \frac{1}{t}\sum_{n=1}^{t} R(s_{t,2/3}) \tag{20}$$

and for $Q^{MF2}$ values it is the average reward obtained in coupled moves:

$$\bar{r}_{t+1} = \frac{1}{t}\sum_{n=1}^{t} [R(s_{t,2}) + R(s_{t,3})] \tag{21}$$

where $R(s)$ is the reward obtained for transitioning into state $s$. This differs from Eldar *et al.* (2020) [31], for which forgetting was to a constant value $\theta$, which was a parameter of the model.

After blocks 2 and 4 the learnt $Q^{MF}$ values are of little use due to the introduced changes to the environment, and, again as in Eldar *et al.* (2020) [31] the agent forgets both the $Q^{MF}$ values and state-transition model $T$ according to Eqs 18 and 19, but with different parameters $\phi'^{MF}$ and $\phi'^{MB}$, respectively.

In addition to on-task replay, the agent also engages in off-task replay immediately before the blocks with changed image-reward associations and spatial re-arrangement. In the former case, the agent uses the new reward function $R'(s)$ that corresponds to the new image-reward associations. In the latter case, the agent generates $\widehat{Q}^{MB}$ values with the new reward function and a state-transition model rearranged according to the instructions albeit with limited success:

$$T \leftarrow (1 - \omega)T + \omega T^{\text{rearranged}} \tag{22}$$

During these two off-task replay bouts, the agent uses the exact same subject-specific parameter values as in on-task replay. The effect of such model re-arrangement on the accuracy of $\widehat{Q}^{MB}$ estimates of the re-arranged environment in MI human subjects, as modelled by our agent, is shown in S6 Fig.

**Initialisations.** $Q^{MF}$ values are initialised to $Q^{MF} \sim \mathcal{N}(\theta, 1)$, and the state-transition model $T(s_{t,1}, m_t, s_{t,2})$ is initialised to 1/7 (since self-transitions are not allowed). The agent, however, starts the main task with extensive training on the same training trials that the subjects underwent before entering the main task. In S8 Fig, we show how ignorant (according to our agent's prediction) each subject was as regards the optimal transitions in the state-space after this extensive training.

## Parameter fitting

To fit the aforementioned free parameters to the subjects' behavioural data we used the Metropolis-Hastings sampling algorithm in the Approximate Bayesian Computation [68] framework (ABC). As a distance (negative log-likelihood) measure for ABC, we took the root-mean-squared deviation between the simulated and the subjects' performance data, measured as the proportion of available reward collected in each epoch. The pseudocode for our fitting procedure is provided in algorithm 1.

**Algorithm 1** Metropolis-Hastings ABC Algorithm for obtaining a point estimate from a mode of the posterior distribution in the parameter space given the initialisation distribution $\mu(\theta)$, data $D$, and a model that simulates the data $\mathcal{M}(D \mid \theta)$. $\theta_t$ denotes the full multivariate parameter sample at iteration $t$.

Set the exponentially decreasing tolerance thresholds $\{\epsilon_t\}_{0,...,T}$ and the perturbation variances $\{\sigma_t^2\}_{0,...,T}$ for $T$ iterations. The perturbation covariances are set as $\Sigma_t = \sigma_t^2 \boldsymbol{I}$, where $I$ is the identity matrix.

```
 1: for iteration t = 0 do
 2:    while ρ(D, D*) > ε₀ do
 3:      Sample θ* from initialisation θ* ∼ μ(θ)
 3:      fro i in {1..5} do
 3:        Simulate data D*ᵢ ∼ M(D | θ*)
 6:      end for
 7:      Calculate average distance metric ρ(D,D*) = ⅕∑⁵ᵢ₌₁ ρ(D,D*ᵢ)
 8:    end while
 9: end for
10: Set θ₀ ← θ*
11: for iteration 1 ≤ t ≤ T do
12:    while ρ(D, D*) > εₜ do
13:      Sample θ* from proposal θ* ∼ N(θₜ₋₁, Σₜ)
14:      if any [θ*]ᵢ is not within support of π([θ]ᵢ) then
15:        continue
16:      end if
17:      for i in {1..5} do
18:        Simulate data D*ᵢ ∼ M(D | θ*)
19:       end for
20:      Calculate average distance metric ρ(D,D*) = ⅕∑⁵ᵢ₌₁ ρ(D,D*ᵢ)
21:    end while
22:    Set θₜ ← θ*
23: end for
```

The fitting procedure was performed for 55 iterations with the exponentially decreasing tolerance threshold $\epsilon_t$ ranging between 0.6 and 0.10. For covariance matrices, we used identity multiplied by a scalar variance with the exponential range between 0.5 and 0.02. To avoid spurious parameter samples being accepted, we simulated our model 5 times with each proposed parameter sample and then used the average over these simulations to compute the distance metric.

Note that the initialisation distribution $\mu(\theta)$ is different from a prior distribution $\pi(\theta)$, since it did not play any role in the algorithm acceptance probability (in fact, we only sampled the full multivariate parameter sample from the initialisation distribution only once in the very first iteration). The acceptance probability for the canonical Metropolis-Hastings ABC algorithm is:

$$\alpha = \min\left(1, \frac{\pi(\theta^*)q(\theta_{t-1} \mid \theta^*)}{\pi(\theta_{t-1})q(\theta^* \mid \theta_{t-1})}\right) \text{ if } \rho(D, D^*) < \epsilon_t \text{ and } 0 \text{ otherwise}$$

Notice that if the prior distribution is chosen to be uniform, or uninformative, then the two $\pi$'s cancel out. Furthermore, if one uses a Gaussian (hence symmetric) proposal distribution $q$, then the implied acceptance probability of the Metropolis-Hastings ABC algorithm is:

$$\alpha = 1 \text{ if } \rho(D, D^*) < \epsilon_t \text{ and } 0 \text{ otherwise}$$

which is exactly how our algorithm 1 operates.

We chose uniform initialisation distributions with support between 0 and 1 for $\eta^{MF1}$, $\eta^{MF2}$, $\eta_r^{MF1}$, $\eta_r^{MF2}$, $\eta^{MB}$, $\phi'^{MB}$, $\phi'^{MF}$, $\omega$. The initialisation distributions for on-line forgetting parameters $\phi^{MF}$ and $\phi^{MB}$, as well as for the fraction of the learning rate for opposite transitions, $\rho$, were specified to be beta with $\alpha = 6$ and $\beta = 2$. Parameters $\theta$ and $\gamma_m$ were chosen to be Gaussian with mean 0 and variance 1. Inverse temperature parameters $\beta_1^{MF1}$, $\beta_2^{MF1}$, and $\beta_2^{MF2}$ had gamma priors with location 1 and scale 1. Since our perturbation covariance matrices were identities multiplied by a scalar variance, and the values for $\xi$ were in general very small compared to the other parameters—to allow for a similar perturbation scale for $\xi$ we sampled it from a log-gamma distribution with location $-1$ and scale 0.01, and the following perturbations were also performed in the log-space. Our fitting algorithm therefore learnt $\log_{10} \xi$ rather than $\xi$ directly. Additionally, parameters with bounded support (such as the learning and forgetting rates) were constrained to remain within the support specified by their corresponding prior distributions.

The fitting procedure was fully parallelized thanks to the implemented python MPI module in a freely available python package *astroabc* [69]. The distribution of the resulting fitting errors is shown in S9 Fig.

## Replay analysis

The objective replay benefit for Fig 6E was computed as the total accrued gain for all example 1-move replay events shown in Fig 6D. That is, we used Eq 16, where $Q^{MF/MB}$ were taken to be the true MF Q-values (or the true reward obtainable for each action), and $\pi_{old}$ and $\pi_{new}$ were the policies before and after all replay events, respectively. The average replay benefit in Fig 6F was computed in the same way as above, but the total accrued gain was averaged over all states where replay occurred (or equivalently, where there was a policy change). Importantly, in Fig 6E we only show the objective replay benefit as a result of the 1-move replay events from Fig 6D. For the overall average, in 2-move trials the 2-move replay events were also taken into account, and the total accrued gain after a 2-move trial was computed as the average over the 1-move and 2-move value function improvements.

The subjective replay benefit shown in S5E Fig was computed in the exact same way as described above; however, for $Q^{MF/MB}$ we took the agent's updated MF Q-values after the replay events (or event) shown in S5D Fig. The average objective value function change from S5F Fig was computed in the same way as described above.

To analyse the agent's preference to replay sub-optimal actions, we extracted the number of times the agent replayed sub-optimal and optimal actions at the end of each trial, which is shown in Fig 6G and 6H for the most recent and all other (or 'distal') transitions, respectively. Due to the torus-like design of the state space, some transitions led to the same outcomes (e.g. going 'up' or 'down' when at the top or bottom rows), and we therefore treated these as the same 'experiences'. For instance, if the agent chose an optimal move 'up', then both 'up' and 'down' replays were counted as replays of the most recent optimal transition. In 2-move trials, a move was considered optimal if the whole sequence of moves was optimal. In such case, the replays of this 2-move sequence and the replays of the second move were counted towards the most recent optimal replays. If, however, the first move in a 2-move trial was sub-optimal, the

second move could still have been optimal, and therefore both sub-optimal replays of the first move and optimal replays of the second move were considered in this case.

For all replays described above, we considered action entropy values at which these replays were executed (shown in Fig 6G and 6H, right column). For every 1-move replay, the corresponding action entropy was computed according to Eq 1. For every 2-move replay beginning at state $s_1$ and proceeding with a sequence of actions $m_1, m_2$, the corresponding action entropy was computed as the joint entropy:

$$\mathbb{H}(s_1, m_1, m_2) = \mathop{\mathbb{E}}_{s_2 \sim T(s_2|s_1,m_1), s_3 \sim T(s_3|s_2,m_2)} [-\log_2 \{T(s_2 \mid s_1, m_1) \cdot T(s_3 \mid s_2, m_2)\}] \quad (23)$$

where in $T(s_2|s_1, m_1)$ the probability of transitioning from $s_2$ to $s_1$ was set to zero; similarly, in $T(s_3|s_2, m_2)$ the probabilities of transitioning from $s_3$ to $s_2$ and from $s_2$ to $s_1$ were set to zero (since back-tracking was not allowed), and the transition matrix was re-normalized as in Eq 11.

### Example simulations

The contour plot in Fig 1D was generated by simulating the agent in 1-move trials in the main state-space on a regularly spaced grid of 150 $\phi^{MF}$ and $\phi^{MB}$ values ranging from 0 to 0.5. For each combination of $\phi^{MF}$ and $\phi^{MB}$, the simulation was run 20 times for 300 trials (in each simulation the same sequence of randomly-generated starting states was used). Performance (proportion of available reward collected) in the last 100 trials of each simulation was averaged within and then across the simulations for the same combination of $\phi^{MF}$ and $\phi^{MB}$ to obtain a single point from the contour plot shown in Fig 1D. The final matrix was then smoothed with a 2D Gaussian kernel of std. 3.5. The model parameters used in these simulations are listed below:

| | |
|---|---|
| $\eta^{MF1}$ | 0.615 |
| $\eta_r^{MF1}$ | 0.615 |
| $\beta_1^{MF1}$ | 1.152 |
| $\eta^{MB}$ | 0.98 |
| $\rho$ | 0.70 |
| $\gamma_m$ | $[-1.105, 1.088, 0, 0.017]$ |
| $\theta$ | 0.088 |
| $\xi$ | 0.0001 |

The data from Figs 1B, 4 and 5, S1A and S1B Fig were obtained by simulating the agent in a simple environment with only two actions available; in each trial the agent had to choose between the two options for 100 trials in total. The agent received a reward of 1 for the sub-optimal action and 10 for the optimal action (except for S7 Fig, there the agent was simulated with reward values that ranged from 0 to 10 with linear increments of 0.1). The following parameters were used:

| | |
|---|---|
| $\eta^{MF1}$ | 0.20 |
| $\eta_r^{MF1}$ | 0.20 |
| $\beta_1^{MF1}$ | 0.90 |
| $\eta^{MB}$ | 0.98 |
| $\phi^{MB}$ | 0.10 |
| $\phi^{MF}$ | 0.20 |
| $\xi$ | 0.00001 |

$Q^{MF}$ values were initialised to 0, the state-transition model was initialised to 0.5, and the bias parameter $\gamma_m$ was set to 0 for each move.

The data from Fig 3 were obtained by simulating multiple agents in the same 2-action environment as described above. All agents were simulated 50 times in 150 trials (data shown only for first 60 trials), and Fig 3 shows average values over those 50 simulations along with the estimated confidence intervals. The following parameters were used:

| | |
|---|---|
| $\eta^{MF1}$ | 0.30 |
| $\eta_r^{MF1}$ | 0.70 |
| $\beta_1^{MF1}$ | 1.00 |
| $\eta^{MB}$ | 1.00 |
| $\phi^{MB}$ | 0.50 |
| $\phi^{MF}$ | 0.50 |
| $\xi$ | 0.001 |

All agents used the same parameters, wherever applicable (i.e., agents without replay had the relevant parameters from those listed above; agents without forgetting had $\phi^{MF}$ and $\phi^{MB}$ set to 0).

The estimated gain from Fig 4A was computed by using Eq 16 where MF $Q$-values were taken to be the agent's MF $Q$-values after the final trial, and 700 $\widehat{Q}^{MB}$ values were manually generated on the interval $[-9, 12]$. This procedure was repeated with the agent's MF $Q$-values additionally decayed according to Eq 18 with $\phi^{MF}$ values of 0.3, 0.5 and 0.7, and the average reward obtained by the agent over those 100 trials. The x-axis was then limited to the appropriate range.

The estimated gain from Fig 5B was computed in the same way as described above, but instead of manually generating $\widehat{Q}^{MB}$ values, 100 transition probabilities (with constant linear increments) were generated on the interval from $[0.5, 1]$ for the correct transition and $[0, 0.5]$ for the other transition (such that the two always summed up to 1). These multiple instances of the state-transition model were used to generate $\widehat{Q}^{MB}$ values for computing the estimated gain.

The entropy range difference in S1A Fig was computed as the difference between the sub-optimal and optimal action entropy values at which the estimated gain became positive for each respective action after the last (100th) trial. These range differences for each combination of the reward values were then averaged over 20 simulations and plotted as a contour plot which was additionally smoothed with a 2D Gaussian kernel of 1.5 std.

The pessimism bias in S1B Fig was computed as the average difference between the number of sub-optimal and optimal replays at the end of each trial, averaged over the same 20 simulations. The resulting matrix was also smoothed with a 2D Gaussian kernel of 1.5 std.

The contour plot in S1C Fig was generated from the data obtained from the same simulation as for Fig 1D, and using the procedure outlined above (the bias for each combination of $\phi^{MF}$ and $\phi^{MB}$ values was averaged over the last 100 trials and then across 20 simulations). The resulting matrix was smoothed with a 2D Gaussian kernel of 3.5 std.

## Supporting information

**S1 Fig. Replay entropy range and pessimism bias.** (A) Difference in the range of sub-optimal and optimal action entropy values at which the estimated gain for the corresponding actions was positive. Higher values correspond to a higher action entropy range for the sub-optimal action; the agent can therefore benefit from the replay of sub-optimal actions at a wider range of action entropy values. Each datum is an average over 20 simulations where each simulation

consisted of 100 trials. (B) Pessimism bias, defined here as the average (over 100 simulation trials) difference between the per-trial average number of replays of sub-optimal and optimal actions. Each datum is an average over the same 20 simulations. Negative values indicate a stronger pessimism bias. Note that both matrices are triangular, since the reward for a sub-optimal action has to be smaller than that for the optimal action (by definition); the labels on the diagonal therefore label rows. (C) Same quantity as in (B) but for the agent simulated in the behavioural task with varying MF and MB forgetting parameters.
(EPS)

**S2 Fig. Comparison of forgetting parameters.** (A) Comparison of our best-fitting estimates of the MF forgetting parameter, $\phi^{MF}$, to that of Eldar *et al.* (2020) [31] for the hybrid MF/MB model. Our modelling predicted the subjects to be notably more forgetful as regards their MF $Q$-values (2-sample 2-tailed t test, $t = 6.73$, $p = 2.49 \cdot 10^{-9}$). (B) Same as before but for the MB forgetting parameter, $\phi^{MB}$. By contrast to MF forgetting, we predicted that most of the subjects remembered their state-transition models better (2-sample 2-tailed t test, $t = -3.92$, $p = 0.0009$). *** $p < 0.001$.
(EPS)

**S3 Fig. Parameter equivalents of subjects' decision flexibility.** (A) Linear regression of behavioural individual flexibility (IF) for all subjects, as measured by Eldar *et al.* (2020) [31], against IF (averaged over 100 simulations) computed from our agent with subject-specific parameters (Pearson's coefficient of determination, $R^2 = 0.73$, $p \ll 0.0001$). Subjects for which our model predicted sufficient replay (MI subjects) are highlighted in red, and MF subjects are shown in blue. Further, MI subjects who were found to mostly hurt themselves by replay ($n = 2$) are shown in green. IF of MI subjects, as predicted by our agent, was significantly higher than that of MF subjects (permutation test, $\bar{t} = 0.22.$, $p \ll 0.0001$). (B) Multiple regression of all subjects' best-fitting decision, learning and memory parameters against their respective IF values as predicted by our agent (due to our fitting procedure, we excluded all MB-relevant parameters for MF subjects that did not engage in replay). Vertical black lines show 95% confidence intervals. (C) MF forgetting parameter of MI subjects was found to be significantly higher that that of MF subjects (Wilcoxon rank-sum test, $W = 3.08$, $p = 0.002$). (D) Distribution of MB forgetting parameter, $\phi^{MB}$, in MI subjects. (E) Best-fitting gain threshold statistics across MI subjects. (F) Same as in (B) for MI subjects but regressed against the average policy change for each subject due to replay (shown in Fig 7C). All regression models from (B) and (F) were selected using 5-fold cross-validation with recursive feature elimination (with a minimum of 5 features). * $p < 0.05$, ** $p < 0.01$, *** $p < 0.001$, t test (unless specified otherwise).
(EPS)

**S4 Fig. Epistemology of replay: Another example.** The layout of the figure is similar to that of Fig 6. (A) An example move predicted by our agent with parameters fit to an MI subject. (B) State-transition knowledge of the agent after executing the move in (A). Note how the agent is less ignorant about the transitions compared to that in Fig 6. (C) MF and MB knowledge of the agent. The agent chose a sub-optimal move because its MF $Q$-values had been forgotten to an extent that made the agent believe it was optimal. Online learning (green shading) indeed helped the agent to remember that the chosen move was worse than predicted. (D) Replays executed by the agent after learning online about the move in (A). Note that, despite the agent's accurate MF knowledge, it still chooses to engage in replay. This is because its state-transition model was exceptionally accurate; moreover, our parameter estimates indicated that this MI subject had a very low gain threshold. (E) Changes in the objective value function as a result of the replay in (D). (F) Overall, we found that this MI subject, as predicted by our

agent, achieved an average objective value function improvement of 0.06 reward points as a result of replay (1-sample 2-tailed t test, $t = 2.16$, $p = 0.03$). (G-H) Same as Fig 6. $^*$ $p < 0.05$, $^{**}$ $p < 0.01$, $^{***}$ $p < 0.001$, ns: not significant.
(EPS)

**S5 Fig. How replay can hurt.** The layout of the figure is similar to that of Fig 6. (A) Actual move in a 1-move trial chosen by an MI subject that was also predicted by our agent with subject-specific parameters. (B) State-transition knowledge of the agent after executing the move in (A). Note the agent's extreme ignorance about all transitions. (C) MF and MB knowledge of the agent. The agent chose a sub-optimal move because it s MF $Q$-values (including those for other moves) had been forgotten to an extent that made the agent more likely to choose that move. Note that, according to the agent's MF $Q$-values, the subjectively optimal move (which the agent did not choose) was in fact objectively sub-optimal. Online learning (green shading) did not improve the situation, since the chosen sub-optimal move was found to be more rewarding than what the agent's MF $Q$-values had indicated. The agent's MF and MB knowledge after executing this move, therefore, still incorrectly indicated that the two most rewarding moves were 'down' and 'up' (as opposed to the objectively optimal move 'left'). (D) Replays executed by the agent after learning online about the move in (A). The agent chose to replay all other moves at the state where the trial began, and such replay further exacerbated the agent's knowledge, for it decreased the MF $Q$-value of the objectively optimal action (towards its model's estimate, $\widehat{Q}^{MB}$, shown in pink bar). Such replay decreased the agent's objective value function (as measured with respect to the true obtainable reward) of that state by 0.03 reward points (E). According to the agent's estimate, however, it's subjective value function of that state increased by 0.5 reward points. Note that the replay at other states (e.g., 'spanner' and 'house') was also harmful because of the similar confusion of the agent as regards the objectively optimal actions. (F) Overall, we found that this MI subject, as predicted by our agent, mostly hurt himself by replaying at the end of each trial (average change in objective value function was decreased by 0.075 reward points, 1-sample 2-tailed t test, $t = -3.26$, $p = 0.001$). $^{**}$ $p < 0.01$.
(EPS)

**S6 Fig. Effect of state-transition model re-arrangement.** (A) Root-mean-square deviation between the MI subjects' state-transition model (as predicted by our agent) reward estimates, $\widehat{Q}^{MB}$, and the true reward, $Q^{MF}_{true}$, of each transition in the final block of the task before and after the model was re-arranged as in Eq 22. The RMSD measure shows that the re-arrangement (even with limited success) made the state-transition model's predictions significantly more accurate (2-sample 2-tailed t test, $t = -4.48$, $p = 6.61 \cdot 10^{-5}$). (B) Linear regression of the difference between the two quantities in (A) (before—after) against the ignorance about optimal transitions (measured as the average action entropy across objectively optimal actions). MI subjects with lower state-transition model entropy had significantly more accurate $\widehat{Q}^{MB}$ estimates as a result of the model re-arrangement (Pearson's coefficient of determination, $R^2 = 0.95$, $p \ll 0.0001$.). (C) Simulated IF for MI subjects showed significant correlation to the RMSD measure (Pearson's coefficient of determination, $R^2 = 0.91$, $p \ll 0.0001$). (D) Additionally, we found that the MI subjects' ignorance about objectively optimal action outcomes after the model re-arrangement correlated significantly with the simulated IF (Pearson's coefficient of determination, $R^2 = 0.81$, $p = \ll 0.0001$). This means that uncertainty in the subjects' state-transition model was a good predictor of how well they adapted to the spatial re-arrangement that took place. $^{***}$ $p < 0.001$.
(EPS)

**S7 Fig. On-line replay statistics with obliviated knowledge.** The layout of the figure is identical to that of Fig 7. Before engaging in off-task replay prior to blocks 3 and 5, each agent's MF $Q$-values were zeroed-out to examine whether the overall on-line replay statistics would remain unaltered—and indeed they did. Wilcoxon rank-sum test in (A) and (B), 1-sample 2-tailed t test in (C) and (D). $^*$ $p < 0.05$, $^{**}$ $p < 0.02$, $^{***}$ $p < 0.001$.
(EPS)

**S8 Fig. Entropy of pre-trained state-transition models.** We assessed how ignorant each subject (as modelled by our agent) was about optimal 1-move transitions after learning in the training trials and immediately before entering the main task. The ignorance was computed as the average action entropy (Eq 1) across all objectively optimal transitions. MI subjects who, according to our modelling, engaged in replay ($n = 20$) were significantly less ignorant about objectively optimal transitions compared to MF subjects (2-sample 2-tailed t test, $t = -2.56$, $p = 0.014$). The horizontal dashed black line shows maximum ignorance. $^{**}$ $p < 0.01$.
(EPS)

**S9 Fig. Fitting errors.** Distribution of the final fitting errors (RMSD of the proportion of available reward collected between the human subjects and the agent).
(EPS)

## Author Contributions

**Conceptualization:** Georgy Antonov, Christopher Gagne, Peter Dayan.

**Formal analysis:** Georgy Antonov.

**Funding acquisition:** Peter Dayan.

**Investigation:** Georgy Antonov.

**Methodology:** Georgy Antonov.

**Project administration:** Peter Dayan.

**Resources:** Eran Eldar, Peter Dayan.

**Software:** Georgy Antonov.

**Supervision:** Christopher Gagne, Peter Dayan.

**Validation:** Georgy Antonov, Christopher Gagne, Peter Dayan.

**Visualization:** Georgy Antonov.

**Writing – original draft:** Georgy Antonov, Christopher Gagne, Peter Dayan.

**Writing – review & editing:** Georgy Antonov, Christopher Gagne, Eran Eldar, Peter Dayan.

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
