## [Decision Letter · Decision Letter 0]

2 Aug 2021

Dear Mr. Antonov,

Thank you very much for submitting your manuscript "Optimism and pessimism in optimised replay" for consideration at PLOS Computational Biology.

As with all papers reviewed by the journal, your manuscript was reviewed by members of the editorial board and by several independent reviewers. In light of the reviews (below this email), we would like to invite the resubmission of a significantly-revised version that takes into account the reviewers' comments.

In particular, the authors should ensure that this manuscript is presented in a straightforward and comprehensible manner, so that it can be understood by a wide audience in both theoretical and empirical neuroscience; and that the assumptions made and their wider implications for our understanding of replay and memory function are clearly described.

We cannot make any decision about publication until we have seen the revised manuscript and your response to the reviewers' comments. Your revised manuscript is also likely to be sent to reviewers for further evaluation.

Sincerely,

Daniel Bush

Associate Editor

PLOS Computational Biology

Daniele Marinazzo

Deputy Editor

PLOS Computational Biology

Reviewer's Responses to Questions

**Comments to the Authors:**

Reviewer #1: Thank you for the opportunity to review this article. I have been interested in an article like this since seeing the Carey et. al (2019) paper as I thought it would be interesting to see how consistent / inconsistent the Mattar & Daw (2018) model would be with those results.

The authors conduct an impressive modeling analysis of pessimized replay in a (previously published) human behavioral dataset, arguing that pessimized replay is indeed optimal in the face of forgetfulness.

My main critique has to do with some of the (potentially incorrect) assumptions in the article. I do think they can be addressed and thus hope to see this article published after clarification on these issues.

There are two critiques here. Upon further deliberation, I think they are intimately related, but I will sketch them out separately and then try to offer a unified account.

1. It's not clear to me that pessimized replay is inconsistent with the Mattar & Daw model as is (which the present authors discuss, albeit briefly, in lines 214-223). Consider the following case, in which there are 2 actions, L and R. The MB values are 1 and 10, respectively, whereas the MF values are 2 and 10. There is higher Gain (and thus EVB) when replaying the pessimistic action. Thus, replay can be used to devalue options, which was my interpretation of the Carey et. al (2019) finding. I consider this hypothesis to be simpler than the authors' forgetting hypothesis, and thus would like to see it ruled out first before considering more complex hypotheses.

2. Second, I'm not sure I understand the authors idea of MF forgetting. From what I do understand, the problem the authors are identifying is the following. Since TD learning operates under the assumption that the optimal policy will be chosen after the first step, once the optimal policy is reached via training, the value (and thus the probability, assuming a softmax policy) of suboptimal actions will be inflated because the agent will make good decisions afterwards.

The authors say "Insofar as the agent improves over the course of the task, the average reward it obtains increases with each trial. MF values for sub-optimal actions, therefore, tend to rise towards this average experienced reward; MF Q-values for optimal actions, on the other hand, become devalued, as the agent is prone occasionally to choose sub-optimal actions due to its non-deterministic policy. In other words, because of MF forgetting, the agent forgets how good the optimal actions are and how bad the sub-optimal actions are."

Is the claim that this optimistic estimate of MF values is considered 'MF Forgetting'? This seems to be a different definition of 'forgetting' than what is commonly thought.

OR

Are the authors claiming that changing the temperature in order to inject more indeterminism into the policy is forgetting? This, I think, would be a more palatable interpretation of forgetting, but the rationale for this is lost on me, especially given that it runs counter to the concept of annealing.

My attempt at reconciliation in order to understand the authors' intuition (but this is surely to be incorrect, so please correct me) is that the authors have identified that the value of suboptimal actions becomes inflated due to the mathematical formalism of TD learning and thus replay may function as a mechanism to devalue these actions to prevent further use (and thus we see pessimistic replay). If this is indeed what is happening, I worry about using term 'forgetting' so prominently as I believe it has very different connotations than something like 'devaluation'. I understand I may be harping on terminology here, so I do apologize; I do think it's an important distinction, though.

If I am incorrect in understanding the authors' intuition, please correct my mistakes.

Regardless, this was a very impressive modeling paper, and I am grateful to have had the opportunity to review it. I hope my comments were constructive.

Reviewer #2: In this manuscript, Antonov and colleagues investigates the effect of forgetting in a DYNA agent that selects optimal experiences for replay. Through various simulations and model fits to subject behavioral data, the authors explore two types of forgetting: the forgetting of action values, and the forgetting of action outcomes. They find that, under a set of modeling assumptions, a forgetful DYNA agent will tend to prioritize the replay of suboptimal actions and not of optimal actions. They then re-interpret the findings of a previous MEG study to suggest that subjects, who exhibited substantial forgetting, might have been replaying optimally when replaying suboptimal actions.

The topic addressed is timely, given the large number of recent studies investigating the content and function of replay in rodents and humans. In particular, the elegant modeling exercise presented in the article might offer new light into existing replay data that, at first blush, may have appeared inconsistent with a normative view. However, while the authors clearly conducted a careful analysis of how forgetting might affect optimal replay, the presentation of their findings in the paper was rather confusing. Accordingly, below I present a few questions, comments, and suggestions that might improve the clarity of the paper to a broad audience.

MAJOR COMMENTS:

1. First and foremost, the paper uses many jargons and abbreviations. Please consider simplifying the language and reducing the number of abbreviations.

2. The paper presents an enourmous amount of information from the very beginning of the Results section. I suspect that any reader not familiar with both Eldar et al (2020) and Mattar & Daw (2018) would have difficulty understanding the model. This might be mitigated with a clearer and more organized structure. One possibility is to organize the simulations in increasing levels of complexity, such as the following: (i) no replay; (ii) no replay with forgetting of values; (iii) normative replay with no forgetting; (iv) normative replay with forgetting of values; (v) normative replay with forgetting of the transition function (i.e. where the discussion action entropy would fall); (vi) normative replay with both forgetting of value and of the transition function. A structure such as this might help the reader understand how each factor, individually, contributes a different pattern reflected in behavior. Other structures are also possible, but the general aim should be to introduce the model components slowly such that any naive reader is able to understand it.

3. Forgetting is modeled as (i) MF values decaying towards a global average; and (ii) Transitions becoming more uniform. My understanding was that, in the absense of forgetting, the proposed model would reduce to the optimal model. Is that correct? If so, it would beneficial to state that explicitly in the paper. If not, it would be important to explain why.

4. What would be the effect of considering also the forgetting of the (stimulus-specific) rewards?

5. Similarly, how important is the assumption that MF values decay towards a global average? Would the simulation results be different if the decay was, instead, towards a subject-specific baseline?

6. I confess that I did not understand the selection of the very peculiar form for the the softmax in 2-step choices (equation [3]). What is the reasoning behind using both Q-values in 2-move trials, instead of only the Q-value for 2-move sequences (which, according to [8], should converge to the total reward obtained in 2 steps)? How different would the simulation results be if only Q^{MF2} was used? Similarly, why is Q^{MF2} used in equation [5], instead of only Q^{MF1}? How dependent are the simulation results on this assumption?

7. In equation [13], the authors assume that Q^{MB} values for 2-move sequences are calculated on-policy. Why not assume optimal behavior at the 2nd action (i.e. using a max operator instead of pi in [13]), and how important is this assumption?

8. In line 781, the author state that the online forgetting parameters used a beta prior with alpha=6 and beta=2, which has a mean of 0.75 and a standard deviation of only 0.14. I found this choice of prior to be rather specific, contrasting sharply with the more uninformative choices for all other priors. I would have loves to see how the simulations would have turned out if a uniform prior had been used. Would subjects still have been found to be forgetful?

MINOR COMMENTS:

9. Line 192-196: "we find that at high MF forgetting, replay confers a noticeable performance advantage to the agent provided that MB forgetting is mild (as can be seen from the curvative of the contour lines in Fig 2D)". However, Fig 2D only compares replay models with different forgetting rates. Does replay really confer a performance advantage in comparison to an agent that performs no replay? In other words, does replay with substantial forgetting hurt performance, and why/how?

10. Fig 3A is extremely difficult to parse. Same for Fig 3B (e.g. what do the axes mean?).

11. Avoid using asterisks in actions (as in equation [4]), which in RL is generally used to indicate an optimal action.

12. Similarly, in equation [4] and others, it was not clear why Q^{MF2} sometimes had three arguments and sometimes two.

13. R(s) is used to indicate a reward model in equations [12-13], and a sampled reward in [7-8]. Consider using different notation in each case.

14. On line 782, the authors state: "All the other parameters had gamma priors with location 1 and scale 1". Please list all such parameters explicitly.

**Have the authors made all data and (if applicable) computational code underlying the findings in their manuscript fully available?**

Reviewer #1: Yes

Reviewer #2: Yes

PLOS authors have the option to publish the peer review history of their article (what does this mean?). If published, this will include your full peer review and any attached files.

Reviewer #1: No

Reviewer #2: No
---

## [Decision Letter · Decision Letter 1]

25 Oct 2021

Dear Mr. Antonov,

Thank you very much for submitting your manuscript "Optimism and pessimism in optimised replay" for consideration at PLOS Computational Biology. As with all papers reviewed by the journal, your manuscript was reviewed by members of the editorial board and by several independent reviewers. The reviewers appreciated the attention to an important topic. Based on the reviews, we are likely to accept this manuscript for publication, providing that you modify the manuscript according to the review recommendations. Specifically, if you could incorporate some additional details regarding your definition of 'forgetting', as specified by Reviewer 1.

Sincerely,

Daniel Bush

Associate Editor

PLOS Computational Biology

Daniele Marinazzo

Deputy Editor

PLOS Computational Biology

[LINK]

Reviewer's Responses to Questions

**Comments to the Authors:**

Reviewer #1: I thank the authors for addressing my comments and for a comprehensive revision.

I believe most of the confusion was due to the prominent use of 'forgetting' in a context I was unfamiliar with. I would ask the authors to

(1) cite other work that that uses this definition of forgetting (i.e. decay to average reward rate) since they claim it is the standard and

(2) comment on whether this is a proposal for what forgetting is defined as in a psychological context, or if this is specific jargon used in technical reinforcement learning settings that does not necessarily bear on (folk) psychological notions of forgetting

Reviewer #2: The structure and clarity of the paper has greatly improved. Thank you very much for taking the time to implement these changes. I am sure that this will increase the impact of the paper.

I have only a few final minor suggestions left:

* Line 111: The caption says "Unfilled hexagons show epochs which contained trials without feedback", but the figure shows no unfilled hexagons.

* Fig 3B: Please add the legend indicating the meaning of the yellow and blue lines (currently, there's only a label for the dashed lines)

* Line 237: In the previous sentence (line 235-236), the authors say "provided that MB forgetting is mild", followed by "This means that despite moderate uncertainty in the transition structure, the agent is still able to improve its MF policy and increase the obtained reward rate." Don't the authors mean: "as long as there is little uncertainty in the transition (...)"?

**Have the authors made all data and (if applicable) computational code underlying the findings in their manuscript fully available?**

Reviewer #1: Yes

Reviewer #2: Yes

PLOS authors have the option to publish the peer review history of their article (what does this mean?). If published, this will include your full peer review and any attached files.

Reviewer #1: No

Reviewer #2: No

Figure Files:

Data Requirements:

Reproducibility:

References:

---

## [Editor Report · Decision Letter 2]

12 Nov 2021

Dear Mr. Antonov,

We are pleased to inform you that your manuscript 'Optimism and pessimism in optimised replay' has been provisionally accepted for publication in PLOS Computational Biology.

Best regards,

Daniel Bush

Associate Editor

PLOS Computational Biology

Daniele Marinazzo

Deputy Editor

PLOS Computational Biology

---

## [Editor Report · Acceptance letter]

20 Dec 2021

PCOMPBIOL-D-21-01256R2 

Optimism and pessimism in optimised replay

Dear Dr Antonov,

I am pleased to inform you that your manuscript has been formally accepted for publication in PLOS Computational Biology. Your manuscript is now with our production department and you will be notified of the publication date in due course.

With kind regards,

Zsofia Freund
